



# Temperate Oligocene surface ocean conditions offshore Cape Adare, Ross Sea, Antarctica

Frida S. Hoem[1], Luis Valero[2], Dimitris Evangelinos[3], Carlota Escutia[3], Bella Duncan[4], Robert M. McKay[4], Henk Brinkhuis[1,5], Francesca Sangiorgi[1], and Peter K. Bijl[1]

[1]Department of Earth Sciences, Utrecht University, Utrecht, The Netherlands, [2]Department of Earth Sciences, University of Geneva, Geneva, Switzerland, [3]Instituto Andaluz de Ciencias de la Tierra (CSIC-UGR), Armilla, Spain, [4]Antarctic Research Centre, Victoria University of Wellington, Wellington, New Zealand, [5]Royal Netherlands Institute for Sea Research (NIOZ), Texel, The Netherlands

*Correspondence to*: Frida S. Hoem (f.s.hoem@uu.nl)

**Abstract**. Antarctic continental ice masses fluctuated considerably in size during the elevated atmospheric $CO_2$ concentrations (~600–800 ppm) of the Oligocene "coolhouse". To evaluate the role of ocean conditions to the Oligocene ice sheet variability requires understanding of past ocean conditions around the ice sheet. While warm ocean conditions have been reconstructed for the Oligocene Wilkes Land region, questions arise on the geographical extent of that warmth. Currently, we lack data on surface ocean conditions from circum-Antarctic locations, and ice-proximal to ice-distal temperature gradients are poorly documented. In this study, we reconstruct past surface ocean conditions from glaciomarine sediments recovered from the Deep Sea Drilling Project (DSDP) Site 274, offshore the Ross Sea continental margin. This site offshore Cape Adare is ideally located to characterise the Oligocene regional surface ocean conditions, as it is situated between the colder, ice-proximal Ross Sea continental shelf, and the warm-temperate Wilkes Land Margin in the Oligocene. We improve the existing age model of DSDP Site 274 using integrated bio- and magnetostratigraphy. Subsequently, we analyse dinoflagellate cyst assemblages and lipid biomarkers ($TEX_{86}$) to reconstruct surface paleoceanographic conditions during the Oligocene (33.7–25.4 Ma). Both $TEX_{86}$-based sea surface temperature (SST) and microplankton results show temperate (10–17°C ± 5.2°C) surface ocean conditions at Site 274 throughout the Oligocene. Increasingly similar oceanographic conditions between offshore Wilkes Land margin and Cape Adare developed towards the late Oligocene (26.5–25.4 Ma), likely in consequence of the widening of the Tasmanian Gateway, which resulted in more interconnected ocean basins and frontal systems. To maintain marine terminations of terrestrial ice sheets in a proto-Ross Sea with as warm offshore SST as our data suggests, requires a strong ice flux fed by intensive precipitation during colder orbital states in the Antarctic hinterland, but with extensive surface melt of terrestrial ice during warmer orbital states.

## 1. Introduction

The Southern Ocean plays a crucial role in global ocean circulation, stability of the Antarctic ice sheet and the carbon cycle. At present, strong temperature gradients isolate Antarctica from warm influence from lower latitude regions. Despite its crucial role, little is known about the evolution of Southern Ocean conditions. Southern Ocean surface conditions cooled during the middle Eocene (Bijl et al., 2009; 2013), which culminated with the initiation of Antarctic continental-scale glaciation at the Eocene-Oligocene transition (EOT~33.7 Ma; Zachos et al., 1994;



Coxall et al., 2005; Bohaty et al., 2012). The overall higher bedrock elevation and larger subaerial area of Antarctica during the Oligocene (Wilson et al., 2013; Paxman et al., 2019) allowed for the occupation of large terrestrial ice caps, Antarctic ice-proximal records imply these ice sheets extended onto the coast, into marine

terminations (Escutia et al., 2011; Scher et al., 2011; Galeotti et al., 2018). Apparently, Southern Ocean temperatures at the earliest Oligocene isotope stage (Oi-1) cooled sufficiently to sustain the marine-terminating ice sheets. Following the Oi-1, gradually deep-sea $\delta^{18}O$ rebounds (Zachos et al., 2008), suggesting long-term loss of Antarctic ice, and/or gradual deep-sea warming. Indeed, the Oligocene remains a relatively warm time interval globally (O'Brien et al., 2020). Moreover, on orbital time scales, Oligocene Antarctic ice volume underwent major

fluctuations in size (e.g., Pälike et al., 2006; Galleoti et al, 2016; Liebrand et al., 2017; Levy et al., 2019), and as of yet it is poorly understood what role Southern Ocean SST conditions played in these fluctuations.

Warm-temperate Oligocene sea surface temperatures (SSTs, 13–25°C) and frontal system reconstructions at the Wilkes Land margin are derived from dinoflagellate cysts (dinocysts) assemblages at Deep Sea Drilling Project (DSDP) Site 269 (Evangelinos et al., 2020) and Integrated Ocean Drilling Program (IODP) Site U1356 (Bijl et

al., 2018b). These were corroborated with quantitative SST based on organic biomarkers (TEX$_{86}$; Hartman et al., 2018), and sedimentological and lithological interpretations (Salabarnada et al., 2018; Evangelinos et al., 2020). These seem to indicate a southward displacement of the (proto-) Southern Ocean fronts, perhaps favoured by the still constricted Tasmanian Gateway (Scher et al., 2015), and consequent southward deflection of warm ocean currents (Fig. 1b). The relative absence of iceberg-rafted debris in most of the Oligocene sedimentary record of

IODP Site U1356 (Escutia et al., 2011; Salabarnada et al., 2018; Sangiorgi et al., 2018; Passchier et al., 2019) suggests that the East Antarctic Ice Sheet (EAIS) at the Wilkes Land sector may have been predominately land-based, indicating limited ice sheet-ocean interaction in this sector of the EAIS. Sedimentary records recovered from cores located near Transantarctic Mountain outlet glaciers: DSDP Site 270 (Kulhanek et al., 2019); ANtarctic geological DRILLing project (ANDRILL; Harwood et al., 2008; Naish et al., 2009; McKay et al., 2016), CIROS-

1 (Barrett et al., 1989), Cape Roberts Project (CRP) (Naish et al, 2001; Prebble et al., 2006; Houben et al., 2013), have provided important insights into widespread marine based advances of both East and West Antarctic Ice Sheet into the western Ross Sea. TEX$_{86}$-based SST records indicate colder temperatures (6–14°C) in the Ross Sea during the Oligocene (Levy et al., 2016; Duncan, 2017) than offshore the Wilkes Land margin (Sangiorgi et al., 2018; Hartman et al., 2018), suggesting a large (~ 7°C), much larger than present, ocean temperature gradient

between the two sectors. However, it remains unknown whether the warm conditions offshore the Wilkes Land margin were unique to that deep-water continental rise sector of Antarctica, or whether similar temperatures existed close to the Ross Sea continental shelf in the Oligocene.

To this end, we investigated sediments recovered from the deep-water continental rise DSDP Site 274, located

offshore the Ross Sea, ~ 250 km northwest of Cape Adare (Hayes et al., 1975), which is at an intermediate location before the aforementioned sites in the Ross Sea and offshore Wilkes Land (Fig. 1). DSDP Leg 28 retrieved valuable sedimentary records from the continental shelf and rise regions of the Ross Sea, but poor age control has long hampered their use in reconstructing past ocean conditions. Moreover, the archives were devoid of calcareous foraminifers, the classic tools used for the reconstruction of ocean conditions. Studies on dinocysts have allowed

recent progress in both age control and paleoceanographic interpretations, as a result of the strong links between



dinocyst assemblage composition and surface water conditions of present-day Southern Ocean (Prebble et al., 2013). New dinocyst records from the Ross Sea region (notably CRP (Clowes et al., 2016) and DSDP Site 270 (Kulhanek et al., 2019)), and from Wilkes Land (IODP Site U1356 (Sangiorgi et al., 2018; Bijl et al., 2018a, b)) provided new biostratigraphic constraints. We used these, together with new biostratigraphic and

magnetostratigraphic analyses to improve the age model of DSDP Site 274, interpret paleoceanographic conditions with dinocysts assemblages, and generate quantitative SST reconstructions with lipid biomarkers (TEX$_{86}$). By comparing these results with available reconstructions from the Ross Sea and Wilkes Land in selected time slices, we evaluated how surface oceanographic conditions changed and latitudinal heat transport developed through the Oligocene.

## 85     2. Material

### 2.1 Site description

DSDP Site 274 (68°59.81'S; 173°25.64'E; 3326 m water depth, Fig. 1a), is located on the lower continental rise in the northwestern Ross Sea, about 250 km north-northeast of Cape Adare (Hayes, 1975). Sediments were collected using punch core-rotary drilling on the *Glomar Challenger* in February 1973 (Hayes, 1975). Currently,

the region is seasonally covered by sea ice (Fetterer et al., 2020) and present-day mean annual SST is ~ -1°C (Locarnini et al., 2019). The site is in the vicinity, south of the upwelling at the Antarctic Divergence and currently located in the path of a major outflow for Antarctic Bottom Water, spilling out over the Ross Sea continental shelf where it is deflected westward (Orsi and Wiederwohl, 2009). The location of Site 274 is ideal for studying the oceanic properties offshore the Ross Sea (Fig. 1b), which we compare to documented Antarctic ice sheet and

ocean conditions from proximal Ross Sea records (Fig. 1a).

### 2.2 Lithology and depositional settings

Drilling at DSDP Site 274 penetrated 421 meters below the sea floor (mbsf) and recovered a total of 43 cores containing 275.5 meters of sediment. We focus our study on the interval between 174.2 and 408.5 mbsf (Cores 19-43)(Fig. 2a). Sediment within this interval is mainly composed of (i) diatom-rich detrital silty clay with varying

amounts of diatoms from trace amounts to up to 80% (diatom ooze) (174.2–328 mbsf); and (ii) silty claystones and interbedded chert layers (328–408.5 mbsf). Scattered iceberg-rafted debris (IRD; pebbles, granules) have been documented between 152 and 323 mbsf. Below 323 mbsf, chert layers compromised core recovery and at 415 mbsf the basalt basement was reached (Hayes et al., 1975). The sediment cores are rather homogenous and lack strong sedimentary structures. The strong biscuiting and fracturing of lithified sediment testifies drilling

disturbance due to the rough nature of rotary drilling, and may have obscured depositional sedimentary structures. Downslope transport of sediment from the Ross Sea continental shelf to the site potentially complicates the reconstruction of local pelagic-derived ocean conditions. The lithology and the seismic patterns (Hayes et al., 1975) suggest that sediment in the Oligocene was transported and deposited within the Adare Basin through a combination of downslope gravity currents and subsequent reworking by bottom currents (Hayes et al., 1975).



## 3. Methods

### 3.1 Age model

The shipboard age model (Hayes et al., 1975) was based on few biostratigraphic (diatom, radiolarian and calcareous nannofossils) age tie points. More recently, Cande et al. (2000) using paleomagnetic data dated the ocean crust underneath DSDP Site 274 to chron 13, ~33.5 Ma. 200 kyr younger than the EOT, and 5-7 Myrs younger than dated during the expedition (Hayes et al., 1975). Granot et al. (2010) formulated seismic stratigraphic units, and correlated these units onto the Ross Sea continental shelf. The lowermost regional unconformity (328 mbsf) above the basement (Hayes et al., 1975) corresponds to a Ross Sea unconformity (RSU) found in the Northern Basin, RSU6, estimated to be of lower Oligocene age (34–26.5 Ma) (De Santis et al., 1995; Granot et al., 2010; Kulhanek et al., 2019). The major unconformity at 180.5 mbsf, between Cores 19 and 20 (Hayes et al., 1975) is tied to seismic reflectors RSU4 and RSU4a, aged middle Miocene (18.34–16.5 Ma; Brancolini et al., 1995; Granot et al., 2010). To further improve the age model, we generated new age tie points based on dinocyst biostratigraphy and magnetostratigraphy to better constrain the age of the sedimentary record (Core 43–17). Dinocyst biostratigraphy follows Bijl et al. (2018a) who reassessed dinocyst species first and last occurrence datums calibrated against the international geological time scale GTS 2012 (Gradstein et al., 2012). Magnetic reversals on the sediment samples were identified through stepwise demagnetization experiments performed using the 2G magnetometer with an inline alternating fields (AF) demagnetiser attached to an automatic sample handler in Fort Hoofddijk (Utrecht University), and the 2G-SRM750 Superconducting Rock Magnetometer housed at the Paleomagnetic Laboratory of Barcelona (CCiTUB-CSIC).

### 3.2 Organic geochemistry

To reconstruct sea surface temperature (SST) we applied the $TEX_{86}$ (TetraEther indeX of 86 carbon atoms) proxy (Schouten et al., 2002), based on the temperature-dependent cyclization of isoprenoidal glycerol dialkyl glycerol tetraethers (GDGTs) produced by thaumarchaeotal membrane lipids. GDGTs were extracted from powdered and freeze-dried sediments using an accelerated solvent extractor. Lipid extracts were then separated into an apolar, ketone and polar fraction by $Al_2O_3$ column chromatography using hexane:DCM (9:1, v:v), hexane:DCM (1:1) and DCM:MeOH (1:1) as respective eluents. Of a synthetic $C_{46}$ (mass-to-charge ratio, m/z = 744) 99 ng GDGT standard was added to the polar fraction, which subsequently was dissolved in hexane:isopropanol (99:1, v/v) to a concentration of ~3 mg ml$^{-1}$ and filtered over a 0.45-μm polytetrafluoroethylene filter. The dissolved polar fractions were injected and analysed by high-performance liquid chromatography–mass spectrometry (HPLC–MS), using double-column separation (Hopmans et al., 2016). GDGT peaks in the HPLC chromatograms were integrated using ChemStation software.

#### 3.2.1. $TEX_{86}$ calibrations

Several calibrations exist to convert $TEX_{86}$ values into SSTs based on modern core–top datasets (Kim et al., 2010). We follow the discussion by Hartman et al. (2018), and used the linear calibration by Kim et al. (2010) to calculate the $TEX_{86}$-SST relations which include the high-latitude core-top values. As we present peak areas of individual GDGTs in the supplements (Table S2), other calibrations can be plotted as well.



### 3.2.2 TEX$_{86}$ overprints and bias

We use ratios of GDGTs as proxies to detect potential overprinting factors that may bias the pelagic signature of the sedimentary GDGTs. The relative contribution of terrestrial GDGT input has been reconstructed using the branched and isoprenoid tetraether (BIT) index (Hopmans et al., 2004). Samples with BIT index values >0.4 may

be biased by soil- and river-derived GDGTs (Bijl et al., 2013). However, we do note that the validity of this proxy for soil organic matter input is questioned, now that it becomes clear that branched GDGTs may also be produced in the marine realm (Peterse et al., 2009; Sinninghe Damsté, 2016), and terrestrial ecosystems that also contain crenarchaeol (Pearson et al., 2004). The methane index (Zhang et al., 2011) flags overprint by sedimentary methanogenic activity, GDGT-2/GDGT-3 ratio (Taylor et al., 2013) signals overprint by archaeal communities

dwelling deeper into the water column and GDGT-0/Crenarchaeol ratio (Blaga et al., 2009; Sinninghe Damsté et al., 2009; Taylor et al., 2013) flags overprint by in situ production of isoprenoidal GDGTs in lakes and rivers, and contribution from Euryarchaeota. The ring index (Zhang et al., 2016), can detect deviations from a pelagic character for the different GDGT 'species' within the TEX$_{86}$ index. Samples which had overprinting values in these biasing indices were marked as unreliable. High-latitude TEX$_{86}$-SST reconstructions are believed to be

skewed towards summer temperatures (Schouten et al., 2013; Ho et al., 2014), but studies around Antarctica, have found archaea to appear most abundant in winter and early spring, with maximum abundances in the subsurface at around 100 m (e.g., Church et al., 2003; Kalanetra et al. 2009; Massana et al. 2009). However, there is a general agreement that TEX$_{86}$ captures the relative SST trend (Richey and Tierney, 2016) remarkably well despite these uncertainties, and this will be our main focus.

## 3.3 Palynology

### 3.3.1 Palynological processing and taxonomy

A total of 50 samples, 2 per core (Core 43–17), were processed for palynology by using palynological processing and analytical procedures of the Laboratory of Palaeobotany and Palynology, published previously (e.g., Bijl et al., 2018a). Freeze-dried or oven-dried sediment was crushed and weighed (on average 10 g, SD: <1 g). A tablet

of a known amount of *Lycopodium clavatum* spores (a marker grain) was added prior to palynological processing to allow for quantification of the absolute number of dinocysts per sample. In order to digest carbonates and silicates, the sediment was treated with 30% HCl overnight first to remove calcium carbonate, 38% HF overnight to digest silicates, 30% HCl was then added to remove fluoride gels, and subsequently centrifuged and decanted. Organic residues were isolated between 250 μm and 10 μm sieve meshes, with the help of an ultrasonic bath to

break down and clear out agglutinated organic particles. Residues were mounted on glass slides using glycerine jelly. Palynomorphs were counted using a Leica DM2500 LED transmitted light optical microscope. While the main focus was on dinocysts, terrestrial palynomorphs and acritarchs were quantified as well, and the presence and relative abundance of other organic remains were noted. Dinocyst taxonomy follows Williams et al. (2017), Clowes et al. (2016) and informal species as presented in Bijl et al. (2018a). Specimens were identified to a species

level when possible. A minimum of 200 identifiable dinocysts were counted per slide at 400x magnification, while the remainder of the slide was scanned at 200x magnification to identify rare taxa not observed during the regular count. Samples with counts of <50 in situ specimens were discarded for qualitative assessment. All slides are logged in the collection of the Laboratory of Palaeobotany and Palynology, Utrecht University.



### 3.3.2 Dinocyst paleoecological affinity

Present-day distribution of dinoflagellates and their cysts depends mostly on surface water temperature, but also on nutrient availability, salinity, bottom water oxygen, primary productivity and sea-ice cover (Dale, 1996; Prebble et al., 2013; Zonneveld et al., 2013). Based on the notion that habitat affinities and feeding strategies of most dinoflagellates can be extrapolated to the fossil assemblages, while keeping in mind that most Oligocene dinocyst species and genera are extant, we can utilize 'deep-time' dinocysts assemblages as a paleoceanographic

proxy (Sluijs et al., 2005; Bijl et al., 2013; Prebble, 2013; Crouch et al., 2014; Egger et al., 2018; Kulhanek et al., 2019). We link modern ecological affinities to Oligocene dinocyst taxa in the Pacific sector of the Southern Ocean, following Prebble et al. (2013) and separate the dinocysts assemblages into Gonyaulacoids (G-cyst) and Protoperidinioid (P-cyst) cysts. In the Southern Ocean, G-cyst generally include phototrophic temperate dinocysts, associated with warm oligotrophic, open water conditions (Prebble et al., 2013). At present, G-cysts

are rare in close proximity of the Antarctic ice sheet (Prebble et al., 2013). An exception is *Impagidinium pallidum* which today is found in Antarctic environments in the vicinity of the polar front (Zonneveld et al., 2013). The extant *Operculodinium* spp., *Pyxidinopsis* spp. *Corrudinium* spp., *Impagidinium* spp. and *Nematosphaeropsis labyrinthus* are absent or represent a minor component of the polar assemblages. P-cysts are generally produced by heterotrophic dinoflagellates and are usually found in nutrient-rich environments: river outlets, upwelling

areas, and sea-ice zones. In the Southern Ocean today, where the Antarctic Divergence upwelling favour a dominance of P-cysts, species such as *Brigantedinium* spp., *Lejeunecysta* spp., and *Selenopemphix* spp. are common (Prebble et al., 2013). *Selenopemphix antarctica* is a species that shows affinity to sea-ice conditions (Zonneveld et al., 2013; Marret et al., 2019).

### 3.3.3 Reworked versus in situ dinocysts

One issue of studying sediment records in the proximity of glaciated margins is separating reworked from in situ species, which is needed for obtaining reliable biostratigraphic constraints and paleoceanographic signals. In turn, quantifying the history of reworked material through time may yield information about the depositional conditions on the Ross ice shelf. In this study, we follow the interpretations of Bijl et al. (2018a) and a priori separated dinocyst species into an assumed reworked and an in situ group (Table 1). We applied statistical analysis to test a

priori assumptions (Bijl et al., 2018a) on in situ or reworked dinocyst species and to quantitatively measure co-variability between environmental variables and palynological data. Our palynological data were analysed using Correspondence analysis (CA), a linear ordination method to explore the differences in assemblages between samples. The palynological data (relative abundance) were plotted in the C2 software program (Juggins, 2007) using square root transformation.

## 4. Results

### 4.1 Revised age model

Based on new dinocyst-based first occurrence (FO) and last occurrence (LO) datums found in the record we provide additional age constraints to the age model upon which we correlate new paleomagnetic reversal results to specific magnetic chrons (Gradstein et al., 2012) (Table 2). Paleomagnetic results are in general of low quality



(Fig. 2b). We interpret this to result from both a low natural remnant magnetization (NRM) intensity (typically between 10-50 A/m$^2$) and the likely growth of iron sulfides during ~50 years storage of the cores, which probably are the cause of magnetic noise as well as the partial isolation of the characteristic component in some samples (Fig. S1; Table S1). Because the low quality of results, we are cautious and only confident in those magnetozones with at least 3 adjacent samples sharing similar polarity values. Most of the samples of the upper part show

reversed polarity, except for five normal magnetozones that we consider reliable. The central part of the site (269.12–214.43 mbsf, in grey Fig. 2b) does not show a definite pattern and consequently was not considered for paleomagnetic correlation. The lower part of the core has a very low recovery and is prone in normal polarity directions.

The presence of marker dinocyst *Malvinia escutiana* (FO = 33.7 Ma; Houben et al., 2011; Houben et al., 2019) in

the lowermost sediment sample (Core 43, 404.66 mbsf) directly overlying the basement, confirms an Early Oligocene age of the lowermost sediment that was also suggested from the age of the underlying ocean crust (Cande et al., 2000). Thus, we correlated the normal magnetozone in Core 43 (400.7 mbsf) with magnetic chron C13n. A few sections above we find the FO of *Stoveracysta ornata* (32.5 Ma) at 396.62 mbsf. The FO of *Operculodinium eirikianum* (31.56 Ma) 352.78 mbsf, the FO of *Corrudinium labradori* (30.92 Ma) at 362.42

mbsf and the LO of *Stoveracysta ornata* (30.8 Ma) is found at 323.6 mbsf. Thus we suggest, the longest normal magnetozone found in Cores 33 and 32 (307.1 mbsf), to correlate with chron C9n. Core 21 (~190.8 mbsf) contain one isolated calcareous nannofossil horizon (Burns, 1975) dominated by *Chiasmolithus altus*; according to Kulhanek et al., (2019) at nearby Site 270, this has its highest occurrence (HO) at 25.44 Ma. Cores 34–20 are included in the diatom *Pyxilla Prolungata* zone (Hayes et al., 1975), which suggests an early Oligocene age (>25

Ma), however the last occurrence of *Pyxilla Prolungata* is discussed to go on until Oligocene – Miocene boundary (23 Ma) (Gombos et al., 1977). Based on these initial report biostratigraphic observations (Hayes et al., 1975), we here correlate the base of normal magnetozone of Core 21 (199.47 mbsf) with the base of chron C8n, and the normal magnetozone of Core 20 (182.6 mbsf) as the second last normal chron of the Oligocene, C7n. A few biostratigraphic constraints, including middle Miocene radiolara species in Core 19 (Hayes et al., 1975) indicate

that the latest Oligocene and Oligocene – Miocene transition is missing in a large hiatus of ~7 Myr between Cores 19 and 20 (181.23 mbsf). We abstain from tying the normal magnetozone of Core 19 to a specific chron, due to the limited biostratigraphic markers. Extrapolating linearly between chrono- and biostratigraphic tie points (Fig. 2b; Table 2) we calculated the sedimentation rate in the Oligocene to be between 1.2– 7.4 cm/kyr.

**4.2 Lipid biomarkers**

Thirty-nine of the 42 samples processed for lipid biomarkers showed no indication of overprints by biasing indices (Fig. S2). The low BIT index value (<0.4 throughout; Fig. S2) suggests low terrestrial organic material influence, relative to marine GDGT production. The normal Ring index values (Fig. S3), with only two outliers, suggests normal pelagic contributions to the sedimentary GDGTs. Thus overall, TEX$_{86}$ values represent an in situ pelagic SST signal. Moreover, the absence of co-variance between TEX$_{86}$ and indices for overprint suggest the high

variability in TEX$_{86}$ also represents a pelagic signal. TEX$_{86}$ values range from 0.44 to 0.55. Using the linear calibration of Kim et al. (2010) (Fig. 4c), SSTs vary between 10–17°C (±5.2°C) throughout the record, with noticeable variability. Below 342 mbsf, reconstructed SSTs are relatively high, and variable (10–16°C). Between



335–248 mbsf SSTs are lower and display lower variability (10–13°C) at the same sample resolution as above. An increase in SST of ~6°C at 248 mbsf marks the onset of a second interval with high variability in SST.

### 4.3 Palynomorphs and dinocyst assemblages

Seven of the 50 samples analysed, all from the top of the studied record (186.66–155.68 mbsf) did not contain sufficient dinocysts and were discarded for our analyses. The remaining 43 samples showed varying abundance of four palynomorph groups: reworked dinocysts, in situ dinocysts, terrestrial palynomorphs and acritarchs (Fig. 4a). The sediments below 352.5 mbsf are dominated by reworked dinocysts, which decrease in abundance above this depth. From 352.5 mbsf to the top of the record, in situ dinocysts constitute the most abundant palynomorph group, followed by acritarchs, which slightly increase upcore. Pollen and spores remain low throughout the entire record (<6%). Furthermore, our palynological samples contain a varying amount of pyritized microfossils and amorphous organic material.

#### 4.3.1 Dinocyst taxonomy

Identification of dinocysts on a species level was possible in most cases (Table S3), however some dinocysts were only defined on a genus level when distinctive features were lacking. *Brigantedinium* spp. includes all round-brown specimens. *Batiacasphaera* spp. includes light, sub- spherical cysts with an angular, apical archeopyle. *Pyxidinopsis* spp. has similar features to *Batiacasphaera,* but is normally smaller, has a thicker, slightly darker wall, and is less folded with a smaller precingular archeopyle. Unidentifiable dinocysts with a smooth, spherical, psilate, hyaline wall and a free, angular- rounded operculum, 5–6 sides, generally found within cyst are hereby named Dinocyst sp. 1. The saphopylic archeopyle of Dinocyst sp.1, resembles that of *Brigantedinium* spp. and *Protoperidinium* spp.

#### 4.3.2 Reworked dinocyst assemblages

The lowermost 60 m of the sediment record, below 352.5 mbsf, yield abundant and diverse dinocysts, that are common in Eocene Southern Ocean sediments (Bijl et al., 2013; Cramwinckel et al., 2020; Crouch et al., 2020) including *Vozzhennikovia apertura, Deflandrea antarctica, Enneadocysta* spp. and *Phthanoperidinium* spp. These species are found throughout the entire record, but their relative abundance decreases upsection. We note good preservation of some of the more delicate dinocysts, which have known biostratigraphic ranges that predate the age of the ocean crust underneath DSDP Site 274, therefore we still regard them to be reworked. Although, we cannot rule out an in situ presence of these typical late Eocene dinocysts in the early Oligocene (Bijl et al., 2018a).

#### 4.3.3 In situ dinocyst assemblages

In the lowermost 15 m of the record, below 390.4 mbsf, the (apparent) in situ assemblage (Fig. 4b) is dominated by protoperidinioid (P-cyst) species Dinocyst sp.1 and *Brigantedinium* spp., indicating high nutrient levels in open ocean settings. Given that *Brigantedinium* spp. has preference for open ocean conditions, often with proximity to upwelling areas, we render it unlikely that it was transported from the continental shelf and reworked. *Brigantedinium* spp. and Dinocyst sp. 1 have not been reported from CRP-3 (Clowes et al., 2016) or the Eocene erratics (Levy and Harwood, 2000) from the Ross Sea area. The good preservation state of the delicate species Dinocyst sp.1 and *Brigantedinium* spp., argues for in situ production. The now-extinct P-cyst species *Malvinia*



*escutiana* occurs throughout the record: its relative abundance increases from the bottom of the record towards its
peak interval from 224 mbsf to the top of the record. The large abundance of *M. escutiana* (30–70% of the total
in-situ assemblages) mainly coincide with lower $TEX_{86}$-SST (10–12°C, Fig. 4b, c). At about 335 mbsf, the
dinocyst assemblages change significantly. Above this depth, oligotrophic (G-cyst) dinocyst species associated
with open, possibly warmer and oligotrophic waters dominate the assemblages. The absence of any signs of
oxidation state changes of the sediments, argues against selective preservation mechanisms as explanation for the
higher abundance of G-cysts (Zonneveld et al., 2010). *Batiacasphaera* spp*., Pyxidinopsis* spp. and *Cerebrocysta*
spp. compose the majority of the G-cysts. *Spiniferites* spp. is relatively abundant (~10–20% of the total in situ
dinocyst counts) in the interbedded chert layers below 352.5 mbsf, coinciding with elevated temperatures, and
again, but less prominent, at 221.4 mbsf ($TEX_{86}$-SST: 14.5°C), while they remain low (<4%) for the rest of the
record. *Operculodinium* spp. is common (10–20%) between 201–221 mbsf where reconstructed temperature are
12–16°C. The highest amount of *Operculodinium* spp. (27%) was found at 239.16 mbsf, where $TEX_{86}$-SST is
16.2°C. *Nematosphaeropsis labyrinthus* is only registered between 361–352 mbsf (green line Fig. 4b), notably in
samples with high temperatures (~15°C). *Impagidinium* spp. remain low (< 7%) in all samples. Throughout the
record, polar water indicative dinocyst species are rare. *Selenopemphix antarctica*, a major component of the
modern Antarctic-coastal assemblages (Zonneveld et al., 2013), is never abundant, and present only in few
samples (between 390.44–333 mbsf, and at 302 mbsf). *I. pallidium*, a dinocyst abundant in polar areas of the
modern ocean (Zonneveld et al., 2013; Marret et al., 2019) has a scattered low presence throughout the record.

### 4.3.4 Terrestrial palynomorphs

The consistently sparse pollen assemblages from DSDP Site 274 suggest a shrubby tundra landscape with low-
growing Nothofagaceae and Podocarpaceae. The offshore and off-path location to the wind patterns from the
continent, constitutes for the low pollen numbers, and we cannot make further interpretation to the terrestrial
ecology.

### 4.4 Correspondence analysis

The CA on our palynological results (Fig. 3, Table S4) resulted in the first 2 axes explaining 46% of the total
variance (31% for axis 1 and 15% for axis 2). Most a priori assumed reworked (purple in Fig. 3) taxa have negative
scores on axis 2 (64%). Those taxa that do not, have generally low total counts or relative abundances (small
circles in Fig. 3). Overall, the species we consider to be definitely in situ (sensu Bijl et al., 2018) have negative
scores on axis 1, and reworked taxa tend to cluster on the positive side of axis 1. Terrestrial palynomorphs (pollen
and spores) plot in the same area as the assumed reworked dinocyst taxa. The overall separation of reworked and
in situ taxa on the first CA axis thus gives confidence in our a priori assumptions in what is in situ and what is
reworked (Table 1).

## 5. Discussion

### 5.1 Updated age model

The age model for DSDP Site 274 is updated by additional biostratigraphic datums and magnetostratigraphy to
correlate sedimentary strata to the geological time scale (Gradstein et al., 2012). The age model of the bottom



(early Oligocene, 33.7 Ma, 404.66 mbsf) and top (late Oligocene, 24.5 Ma, 181.23 mbsf) of the studied interval (408.5–174.2 mbsf) has been improved. However, the few existing age constraints for the middle part (mid Oligocene, 307.1–199.5 mbsf) do not allow a significant improvement of the existing age model for this interval (Fig. 2, Table 2). We acknowledge that although our new constraints have improved the age model for the top and bottom of the study interval, large uncertainties remain, due to moderate recovery, reworked material, weak

NRM intensities (Table S1) and limited occurrence of age-diagnostic microfossils. This means that between tie points, sedimentation rates may vary and hiatuses could be present. We therefore plot the data in the depth domain and we simply indicate the age tie points next to the depth scale (Fig. 2; Fig. 4). Notwithstanding these age model uncertainties, the proxy data we present provides a rare glimpse into early to middle Oligocene surface water conditions over a period of several million years, a period when proxies suggest atmospheric $CO_2$ generally

exceeded 600 ppm (Zhang et al., 2013).

**5.2 Paleotemperature and paleoenvironment in the Oligocene at DSDP Site 274**

Temperature, in situ- and reworked palynomorph results together provide integrated paleoceanographic configurations off the Ross Sea margin during the Oligocene (33.7–24.5 Ma) (Fig. 4). Furthermore, we combine our reconstruction with those available around the East Antarctic margin from the Western Ross Sea and the

Wilkes Land to obtain a regional perspective.

**5.2.1 Surface oceanographic conditions**

Both dinocysts assemblages and $TEX_{86}$- based SST results (Fig. 4b, c) consistently suggest temperate surface-ocean conditions. High variability in the dinocysts- and $TEX_{86}$-SST reconstructions reflects highly dynamic surface-ocean conditions. Although P-cyst species are abundant in the top and bottom of the record, the middle

part of the record shows dominant to high abundance of G-cyst species indicating that oligotrophic conditions occurred at the time (Fig. 4b). The dominant presence of G-cysts imply that upwelling in the Antarctic Divergence was greatly reduced or distal to the site. The stronger fluctuation between oligotrophic and heterotrophic dinocyst species above 265 mbsf, could reflect strong migrations of frontal systems. The absence of typical sea-ice affiliated dinocysts suggests sea ice was absent or the sea ice seasonal coverage was strongly reduced compared

to the present-day. Furthermore, dinocyst assemblages contain marine species, indicative of normal ocean salinities, and therefore we find no indications of large meltwater input. However, if the unidentified Dinocyst sp. 1 (turquoise in Fig. 4b) during the early Oligocene (depth > 335 mbsf) resembles the peridinioid genera *Senegalinium* spp., a group which shows a high tolerance to low surface water salinities (Sluijs et al., 2009), could the region be under the influence of meltwater and/or increased precipitation. The overall abundance of reworked

(Eocene) dinocysts suggests enhanced erosion of marine sediments on the Ross Sea continental shelf, and transport thereof towards the abyssal plain. We cannot deduce from our data whether this was induced by wind-driven transport of surface water, or through density-driven bottom water flow cascading down the continental slope. In general, the Oligocene dinocyst assemblages found at DSDP Site 274, are similar to present-day dinocyst assemblages living between the Subantarctic and Subtropical front, where temperatures range from 0–15°C

(Prebble et al., 2013). This is in line with the high $TEX_{86}$-SSTs (10–17°C), that show much warmer surface waters with lower nutrient levels than today where currently, the core is located in an area with seasonal sea ice cover (Fetterer et al., 2020) and average SSTs ~ -1°C (Locarnini et al., 2019).



### 5.2.2 Oligocene oceanography and climate evolution at DSDP Site 274 in a regional context

The generally warm SSTs throughout the Oligocene also suggest that the recorded high productivity at the site was probably not the result of the influence of cold upwelled waters. Yet, in the early Oligocene (404.66–335.34 mbsf) the relative abundant protoperidinioid dinocysts do indicate high nutrient and fresh surface-water conditions (Fig. 4b). Instead of upwelling, we suggest that strong surface-water mixing stimulated ocean primary productivity at the site, perhaps with additional nutrient sources through melting from the Ross Sea continental margin. Rifting of the Western Ross Sea shelf since 60 Ma (Huerta and Harry, 2007) created thick Eocene sedimentary successions

on the Ross Sea shelf. Glacial-isostatic adjustments as a response of the Antarctic ice sheet build-up (~48–34 Ma) caused reorganisation of shelf sedimentation (Stocchi et al., 2013), notably increases in sedimentation rates due to the accumulation space created by higher sea level and bedrock subsidence in some regions, and erosion due to bedrock uplift at others. Strata drilled at DSDP Site 270 on the Ross Sea continental shelf indicate periods of Early Oligocene glacimarine deposition derived from local ice caps nucleated on elevated highs prior to tectonic

subsidence in that region (De Santis 1999; Kulhanek et al., 2019). Turbid meltwater derived from the margins of these marine terminating ice caps, and from glacio-marine/fluvial systems at the margins of outlet glacier along the Transantarctic Mountain front (Fielding et al., 2000), would also allow for transport via a suspended sediment load or downslope processes towards the continental rise at DSDP Site 274, similar to the Wilkes Land continental rise (Bijl et al., 2018b; Salabarnada et al., 2018). The abundance of reworked late Eocene dinocysts is thus

testament to influence of continental-shelf-derived surface water towards the site, which in turn brings with it nutrients that promote peridinioid dinocysts. This high amount of reworked dinocysts could further argue for a reworked TEX$_{86}$-SST signal. However, the near-shore character of the Eocene reworking would have made the reworked GDGT pool enriched in branched GDGTs, which is not what we see in our record. This argues against the TEX$_{86}$ results being influenced by reworked GDGTs from the Eocene. After ~29 Ma (335 mbsf), the relatively

warm TEX$_{86}$-based SSTs (10–17°C) and abundance of offshore, temperate dinocyst species *Operculodinium* spp., *Spiniferites* spp., and *Nematosphaeropsis labyrinthus* (Fig.5b, c) indicates a long period of temperate conditions with less variations over Site 274. The covariance between the dinocyst species and the SST shows the strong link between SST variability and biotic/oceanographic response. The persistent presence in high abundance of *Malvinia escutiana* in the mid Oligocene (<265 mbsf) is unprecedented, extending the LO of the species to a

younger age than previously reported (Bijl et al., 2018a). Its large numbers seems to suggest the conditions at this site were favourable for this species, and makes it unlikely a reworked signal. The CA plot (Fig. 3) shows that *Malvinia escutiana* co-varies with oligotrophic and temperate dinocyst groups as well as with acritarchs. This suggests that *Malvinia* favours open water and low nutrient conditions, potentially influenced by meltwater input. A conundrum in our data is the increase in the G-cysts groups in the mid-Oligocene: *Batiacasphaera* spp.,

*Pyxidinopsis* spp. and *Cerebrocysta* spp., and a decrease in P-cyst abundances synchronous with declining SST starting at ~29 Ma (335.3 mbsf). At present, these G-cysts are associated with more northerly Subantarctic and Subtropical front zone regions (Prebble et al., 2013), with temperate ocean conditions. Although, in general in our record the temperate dinocysts and lipid biomarkers are consistent, between ~29 Ma and 26.8 Ma (335.3–252.2 mbsf) we note an increase in abundance of warm-affiliated dinocysts taxa while SSTs drop. The warm-affiliated

dinocysts are all G-cysts. We here argue that decreasing nutrient levels cause P-cyst to disappear, which leaves the remaining G-cysts higher in abundance. Therefore, the shift in dinoflagellate assemblage is related to nutrient conditions rather than temperature.





Abundance of transparent chorate acritarchs at Site 274 generally follows warmer SSTs, similarly to Site U1356
offshore Wilkes Land (Bijl et al., 2018b). The CA analysis showed little co-variance of acritarchs and reworked
assemblages, thus suggesting that the acritarchs are in situ. At ~26.5 Ma (239.2 mbsf) the acritarchs peak is
synchronous with a peak in temperate dinocyst species *Operculodinium* spp. Previous studies on Antarctic
proximal records, like on CIROS-1 core (Hannah, 1997) and DSDP Site 270 (Kulhanek et al., 2019), have
associated the presence of acritarchs (predominantly *Cymatiosphaera* spp. and *Leiosphaera* spp) with episodes of
sea ice melting. We did not find abundant *Cymatiosphaera* spp. and *Leiosphaera* spp. This is interpreted as
indicating that the fresh-water influence was reduced at Site 274, compared to sites on the Ross Sea continental
shelf that were more proximal to the glaciated margin.

### 5.3 Broader regional perspective

We compare our Oligocene paleoceanographic reconstructions from DSDP Site 274 with records from off the
Wilkes Land margin (Site U1356 (Hartman et al., 2018; Salabarnada et al., 2018; Bijl et al., 2018a, b; Sangiorgi
et al., 2018)) and the Ross Sea; (Houben et al., 2013; Clowes et al., 2016; Kulhanek et al., 2019; Duncan, 2017)
(Fig. 5). Published $TEX_{86}$ data from Wilkes Land margin (Hartman et al., 2018) and the Ross Sea (Duncan, 2017)
have for this comparison been converted to SSTs using linear calibration of Kim et al. (2010) (calibration error:
± 5.2°C).

### Early Oligocene (32.3–29.2 Ma, 391–335 mbsf)

DSDP Site 274 $TEX_{86}$-SST results suggest for the entire study record slightly lower average temperatures (~4°C)
than at Site U1356, but markedly higher than at ice proximal Ross Sea sites (CIROS-1; Fig. 5c). This observation
is consistent with the position of DSDP Site 274, which is at slightly higher paleo-latitudes compared to Site
U1356, and offshore from the ice proximal Ross Sea sites in the cooler Ross Sea catchment. Indeed, evidence
from the CRP cores in the Ross Sea showed continental-scale ice sheets first expanded towards the Ross Sea
around 32.8 Ma (Galeotti et al., 2016). Prior to 31 Ma (350 mbsf), the SST record from DSDP Site 274 shows
some of its highest temperatures  of the record, while SSTs at Site U1356 were already decreasing. One important
consideration is whether these sites in the Ross Sea and Wilkes Land can be adequately compared into a latitudinal
transect, given that they are separated by the Tasmanian Gateway, an evolving conduit that separates the eastern
Indian and southwestern Pacific oceans. Although an ocean crustal connection through the Tasmanian Gateway
was just established in the Oligocene, the passageway was still quite restricted (Stickley et al., 2004; Bijl et al.,
2013), and a question remains on how well connected these two ocean basins were at this time. Studies of the
paleobathymetry and sedimentary mechanisms in the Southern Ocean through the Cenozoic (e.g., Scher et al.,
2015; Hochmuth et al., 2020) do show the Tasmanian Gateway as well as the Pacific sector of the Southern Ocean
deepen between 34 Ma and 27 Ma, allowing easier through flow and exchange between the different ocean sectors.
The little co-variability between the Adare Basin and Wilkes Land margin, and the disparate SSTs might signal
the disconnection between both sites, perhaps by the still restricted Tasmanian Gateway. While synchronous SST
variability and changes therein between the sectors after 31 Ma suggests connection between the ocean basins, in
line with other studies (Scher et al., 2015), a SST difference between both sectors remains. The abundance of low



nutrient/temperate-affiliated dinocyst taxa (G-cyst) is higher at DSDP Site 274 offshore the Ross Sea than those at the Wilkes Land margin and within the Ross Sea continental shelf, implying that at this offshore location the nutrient input was lower than to the study sites proximal to the continent.


**Mid Oligocene (29.1–26.6 Ma, 333.6 – 239 mbsf)**

In the mid Oligocene, the absolute SST average values disparity between DSDP Site 274, the Ross Sea and Wilkes Land margin is the strongest within our studied interval. Both Wilkes Land margin and the Ross Sea have high P-cyst content (Fig. 5b). Palynomorphs from Ross Sea shelf deposits from Oligocene, dominated by *Lejeunecysta*

spp. and acritarch species *Cymatiosphaera* (CRP: Prebble et al., 2006; Clowes et al., 2016), suggest cooling in the Ross Sea region through this time interval, with varying amounts of fresh water input (Prebble et al., 2006). In contrast, our dinocyst assemblages suggest pelagic, low nutrient, marine conditions while the low numbers of terrestrial palynomorphs point to limited fresh-water or melt-water input at DSDP Site 274. Similar to the Wilkes Land margin SST record, DSDP Site 274 SSTs decrease towards the late Oligocene.

**Late Oligocene (26.5 – ~25.4 Ma, 239-192.7 mbsf)**

The average $TEX_{86}$-based SST results (Fig. 5a) for Site U1356 and Site 274 shows larger (>6°C) variability (Hartman et al., 2018; Hartman et al., 2018). At DSDP Site 274, we can exclude the known non-thermal biases as cause for the strong variability, and therefore also interpret stronger SST variability in the late Oligocene. Noteworthy, in the beginning of this interval at 26.5 Ma (239 mbsf) we see a temperature peak at DSDP Site 274

similar to what was reconstructed at the Wilkes Land margin, related to orbital cyclicity, at ~26.5 Ma (Hartman et al., 2018). This temperature peak coincides with a rapid decrease in the $\delta^{18}O$ isotope records that may be linked to the deglaciation of large parts of the Antarctic ice sheet following a large transient glaciation centered on ~26.8 Ma (Pälike et al., 2006). The increase in abundance of *Operculodinium* spp. at all three sites (DSDP Sites 270, 274 and IODP Site U1356) is a testament to the temperate conditions and/or lower nutrient availability at the time.

The DSDP Site 274 sediment record is virtually barren of palynomorphs <192.7 mbsf (~25.5 Ma), approximately ~1 Ma before the hiatus (181 mbsf) in the record (~24.5 Ma), with the sediments above estimated to be of middle Miocene age (Hayes et al., 1975). Since our SST reconstructions exclude continuous sea ice cover as possible explanation, we interpret that oxic degradation consumed palynomorphs at the sea floor. Three reasons for increased oxygen delivery at the sea floor are proposed; 1. Strengthening of the ACC increased deep ventilation.

This is unlikely given that ocean frontal systems would progressively move northward while the Tasmanian Gateway widens, which would also displace ACC flow northwards, away from the site. 2. Decreased sedimentation rates increased oxygen exposure time of palynomorphs at the sea floor. Although our age model is limited, we see a decrease in sedimentation rates to 1.15 cm/kyr for this time interval. 3. Bottom water formation on the Ross Sea continental margin delivered increased oxygen-rich bottom waters to the site. Heightened

obliquity sensitivity was therefore interpreted to be associated with enhanced oceanic-influence mass balance controls on marine terminating ice sheets, with limited sea ice extent (Levy et al., 2019). Levy et al. (2019) interpreted a prominent increase in the sensitivity of benthic oxygen isotope variations to obliquity forcing (termed "obliquity sensitivity") between 24.5 and 24 Ma, synchronous with the first occurrence of ice-proximal glaciomarine sediments in the DSDP Site 270, disconformities in CRP-2/2A, and a large turnover in Southern

Ocean phytoplankton. The major expansion of the ice sheet close to the Oligocene – Miocene boundary in the Ross Sea (Levy et al., 2019; Kulhanek et al., 2019; Evangelinos et al., in review) argues in favour of Ross Sea



bottom water strengthening, but this also potentially enhanced winnowing at the site, leading to the slow-down of the sedimentation rates above 192.7 mbsf and the formation of the >7 Myr duration hiatus at ~181 mbsf.

**5.4 Implications for ice-proximal conditions, hydrology and ice sheets**

Warm and generally oligotrophic conditions at a relatively proximal location imply that the Southern Ocean oceanography was fundamentally different from modern (e.g., Deppler et al., 2017). Although ice-proximal ocean conditions were likely colder inshore than further offshore in our reconstruction, they remain warm considering their proximity to marine-terminating outlet glaciers and ice caps in the Ross Sea area (De Santis et al 1999; Galeotti et al., 2016; Levy et al., 2019; Kulhanek et al., 2019; Evangelinos et al., in review). Levy et al. (2019)

provided a model for ice-proximal to ice-distal oceanographic conditions in the Ross Sea during the Oligocene. In that model, Transantarctic Mountain outlet glaciers draining the EAIS, or local marine-terminating ice caps in the Ross Sea were particularly affected by the wind-driven, southward advection of warmer subsurface waters onto the Ross Sea shelf, similar to how Circumpolar Deep Water is being transported onto some regions of the continental shelf today (e.g., Shen et al., 2018; Wouters et al., 2015). The subsurface waters in that conceptual

model were indicated as warmer than the overlying low salinity surface waters derived from glacial melts during glacial maxima, but this stratification is broken down during interglacials. Our data support this interpretation with the temperate surface waters over the continental rise of the Ross Sea margin suggesting a well mixed water column, with cool stratified meltwater influences largely restricted to coastal Ross Sea sites of DSDP Site 270, CRP and CIROS-1. Our results imply that intermediate waters at Site 274 could hardly have been warmer than

the surface waters. Having temperate surface waters just offshore the Ross Sea shelf provides another means to deliver this heat that restricted the advance of marine terminating glacial systems into the Ross Sea and Wilkes Land continental shelfs. Pollen assemblages and high surface temperatures at DSDP Site 274, supported by terrestrial palynomorphs found at CRP-2 (Askin and Raine, 2000), suggest climate was warm enough to allow atmospheric melt to be the dominant mass balance control and driver of deglaciation during warm orbital

configurations. In addition, the warm ocean conditions offshore the Antarctic ice sheet could have promoted the hydrological cycle in the Antarctic hinterland, similar to the middle Miocene Climatic Optimum (Feakins et al., 2012). Consequently, enhanced intense precipitation in the Antarctic hinterland could have provided sufficient ice accumulation during cold orbital states to sustain some marine terminations of the predominately terrestrial ice sheets. Hinterland precipitation and glaciation in the warmer-than-present climates of the early to mid

Oligocene was further promoted by high elevation and larger Antarctic landmass size at this time (Paxman et al., 2019). Indeed, General Circulation Models (GCMs) for the ice-free Eocene do suggest enhanced precipitation delivery to the Antarctic continent (e.g., Huber and Caballero, 2011; Baatsen et al., 2018). If in fact the part of the source of that precipitation was the warm Southern Ocean, proximal to the ice sheet, rayleigh distillation was reduced, leading to much less depleted $\delta^{18}O$ of the Oligocene ice sheet than today, and thereby, less enriched $\delta^{18}O$

of sea water. The calculation of ice volumes from benthic foraminiferal oxygen isotope records (e.g., Lear et al., 2000; Bohaty et al., 2012; Liebrand et al., 2017) do incorporate a variety of values for the isotopic composition of Oligocene Antarctic ice sheet. We argue that the warm oceanographic conditions, invoking strong precipitation and possible more local source of precipitation than today, would argue that $\delta^{18}O$ of Antarctic ice was on the less depleted end of previous assumptions, which increases the calculated Antarctic ice volume that was installed

during the Oi-1 isotope stage (Bohaty et al., 2012), and the Antarctic ice volume that fluctuated over the strong



Oligocene orbital cycles in benthic $\delta^{18}O$ (Liebrand et al., 2017). Future isotope-enabled ice sheet modelling should include warm Southern Ocean conditions for realistic estimates of Antarctic ice volume. This may imply an even higher sensitivity of Antarctic ice sheets to orbitally forced climate variability than previously assumed, and assigns a large role of mass balance controlled by surface melt (precession driven top down melting) and

oceanography in ice sheet stability during past warm climates, through both hydrological and basal and surface melt processes.

## 6. Conclusion

We show that temperate ($TEX_{86}$-SST: 10–17°C +/- 5.2°C) and relatively oligotrophic surface ocean conditions prevailed off the Ross Sea margin during the Oligocene. This matches the relatively warm settings recorded off

the Wilkes Land margin, and demonstrates that warm surface waters were proximal to the East Antarctic Ice Sheet in both the Ross Sea and Wilkes Land during the Oligocene. Comparing our warm surface ocean results with colder surface records and evidence of temporary marine termination of ice caps and glaciers in the Ross Sea continental shelf, we show that a strong temperature gradient prevailed from inshore to offshore the Ross Sea margin. We posit that the warm surface ocean conditions near the continental shelf break during the Oligocene

may have promoted increased heat delivery and precipitation transport towards the Antarctic ice sheet that lead to highly variable oscillation of terrestrial ice sheet volumes in the warmer climate state of the Oligocene. During cold orbital phases, enhanced precipitation sustained high ice flux and advance of terrestrial ice sheet and ice caps into shallow marine settings, while during warm orbital configurations of the Oligocene, the heat delivery resulted in widespread summer surface melt, and widespread retreat of the terrestrial ice sheets into the hinterland.


**Acknowledgements**

This work used Deep Sea Drilling Project archived samples and data provided and curated by the International Ocean Discovery Program and its predecessors. This study was funded by NWO polar programme grant number ALW.2016.001. DE acknowledges funding through the Alexander S. Onassis Public Benefit Foundation Ph.D.

research grant: F ZL 016-1/2015-2016. DE and CE acknowledge funding through the Spanish Ministry of Economy, Industry and Competitivity (grant CTM2017-89711-C2-1-P/ CTM2017-89711-C2-2-P), co-funded by the European Union through FEDER funds. We thank Natasja Welters and Giovanni Dammers for technical support.

**Author contributions**

PKB and FS designed the research. PKB, CE and DE collected the samples. CE and DE described the cores. LV collected and analysed paleomagnetic samples. FSH processed samples for palynology and organic geochemistry, FSH and PKB analysed the data and FSH wrote the paper with input from all authors.

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



**Table captions**

**Table 1: List of palynomorphs and their abbreviated codes found in the CA-plot (Figure 5). Assumed in situ and reworked dinoflagellate cyst taxa are assigned to protoperidioid (P-cyst) taxa and gonyaulacoid (G-cyst) taxa.**

**Table 2: Improved age model for the Oligocene of DSDP Site 274 determined by dinocysts biostratigraphy indicators (FO = First occurrence, LO = Last occurrence, HO = Highest occurrence) and paleomagnetic reversals (chrons).**

**Figure captions**

**Figure 1: (a) Ross Sea to Wilkes Land margin bathymetry with present-day locations of**
**DSDP/IODP/CRP/ANDRILL drill sites included in this study (red dots). The new data generated for this study comes from DSDP Site 274, marked by yellow dot. The base map is from Quantarctica GIS package, Norwegian Polar Institute. The insert shows the Antarctic continent and the surrounding oceans (divided by gray dotted lines) to give a broader regional context to the study area. (b) A synthesis of paleoceanographic settings at 27 Ma. The paleogeographic position is generated with G-plates**
**(http://www.gplates.org), based on the global plates geodynamic motion model from Müller et al., (2016). Light grey indicates the continental lithosphere. The inferred ocean currents are drawn after reconstructions by Stickley et al., (2004). TC = Tasman current, PLC = Proto-Leeuuwin Current and ACountC = Antarctic Counter Current. Blue arrows indicate cooler ocean currents and red indicate warmer ocean currents. Relative current strength is indicated by arrow size.**

**Figure 2: (a) Core numbers, core recovery and lithological description of the cores based on the initial DSDP reports (Hayes et al., 1975). (b) Magnetic correlation for Site 274. Inclination values define local magnetic polarity zones. Magnetostratigraphic correlation is firstly guided by new dinocyst constraints, biostratigraphic markers from shipboard report and subsequently by correlation between local polarity zones and the GTS2012 timescale (Gradstein et al., 2012). Low intensity, shifting directions, and low**
**recovery precludes magnetozone identification for some intervals. Characteristic orthoplots showing demagnetization steps is included in Supplementary Figure S1. Arrows indicate age (Ma) biostratigraphic tie points according to the age model described in Table 2. Extrapolations has been made between the age tie points (stippled lines) with sedimentation rates indicated in between. HO = Highest occurrence, LO = Last occurrence, FO = First occurrence.**

**Figure 3: Correspondence analysis (CA) of the dinocyst assemblage data from DSDP Site 274. The size of the points indicates the total relative abundance of the specific species. The abbreviations of the dinocysts species can be found in Table 1. The data were plotted in the C2 software program (Juggins, 2007). The analysis scores are provided as Table S4.**





**Figure 4: Lithological (the legend is the same as Figure 2), palynological and TEX$_{86}$-SST results from DSDP Site 274 plotted against depth. Arrows indicate age (Ma) tie points according to the age model described in Table 2. The dotted gray line indicated the time slices selected for Fig. 5.**

**(a) The cumulative relative abundance of palynomorph groups.**

**(b) The cumulative relative abundance (%) of selected dinocysts groups recorded in the studied interval.**
**Blue tones are P-cysts, red-tones are G-cysts.**

**(c) TEX$_{86}$-based SSTs (Linear calibration, Kim et al., (2010)), calibration error is ± 5.2°C, indicated by black bar in bottom of the plot. The TEX$_{86}$ outliers are marked in red.**

**Figure 5: Synthesis of sea surface temperature and dinocysts assemblage changes between the early (c),**
**mid (b) and late Oligocene (a) in the Ross Sea (CRP, ANDRILL, DSDP Site 270), offshore Cape Adare (This study, DSDP Site 274) and Wilkes Land margin (Site U1356). The pie charts visualize the dinocyst assemblage composition at respective sites (see legend). Dinocyst assemblage data from the Wilkes Land margin, U1356, comes from Bijl et al., (2018a, b) for all panels (a-c). Dinocyst assemblage data from the Ross Sea is gathered from DSDP Site 270 (Kulhanek et al., 2019) for panel a) and from CRP (Houben et**
**al., 2013; Clowes et al., 2016) for panel (b) and (c). The TEX$_{86}$-SST data from Wilkes Land, U1356 comes from Hartman et al., (2018), 35 TEX$_{86}$-data points were used; 7 in (a), 9 in (b) and 19 in (c). In the Ross Sea there is a lack of TEX$_{86}$-SST data from the mid Oligocene, but Duncan (2017) presented unpublished TEX$_{86}$-data from CIROS- (12 TEX$_{86}$-data points), here displayed in panel (c), and from DSDP Site 270, where only one data point matched our mid-early Oligocene time slice in panel (a). All TEX$_{86}$ data have**
**been converted to the SST using linear calibration of Kim et al. (2010) (calibration error: ± 5.2°C). The paleogeographic position is generated with G-plates (http://www.gplates.org), based on the global plates geodynamic motion model from Müller et al., (2016).**

**Supplementary Information**
**Supplementary Table S1: Table with a summary of demagnetization data results. Sample identification, Core location indicating core, section and depth (mbsf), Declination, Inclination, Sample intensity (in A/m2), MAD values and remarks including the steps used for interpretation. Resultant orthoplots are depicted in Fig. S1.**

**Supplementary Table S2: Concentrations of GDGTs at Site 274. All samples and corresponding depths, GDGT peak area values, TEX$_{86}$ (Schouten et al., 2002) and BIT index values (Hopmans et al., 2004), Methane Index (Methzhang) values (Zhang et al., 2011), GDGT2/Crenarchaeol ratios (Weijers et al., 2011), GDGT-0/Crenarchaeol ratios (Blaga et al., 2009) and GDGT-2/GDGT-3 ratios (Taylor et al., 2013), and RING index (Sinninghe Damsté, 2016). SST calibrations from Kim et al., 2010; Kim et al., 2012. SSTK10L**
**= linear calibration of Kim et al. (2010). Discarded samples (OUTLIER=TRUE) with outlier values are based on BIT > 0.4, GDGT2/GDGT3` > 5, `GDGT0/cren` > 2 and `Methzhang` > 0.3.**

**Supplementary Table S3: Total palynomorph assemblage counts DSDP Site 274 cores 43–21.**



**Supplementary Table S4: Correspondence analysis (CA) scores of the dinocysts assemblage data from DSDP Site 274.**

**Supplementary Figure S1: Orthogonal plots of representative samples. Most of the samples used for the correlation show two distinctive directions, both in normal samples and in reversed samples. Inclination**
**values are also indicated. Open plots indicate inclinations (vertical projection). All calculated directions are available in Table S1. Samples were calculated by means of the Paldir and paleomagnetism.org (Koymans et al., 2016) programs.**

**Supplementary Figure S2: Relevant GDGT indices to filter out biased outliers (red crosses) in the generated**
**GDGT data (Table S2), plotted against sample depth (mbsf). The red line marks the limit of reliable values. a) $TEX_{86}$ (Schouten et al., 2002). b) BIT index values (Hopmans et al., 2004). c) Methane Index (Methzhang) values (Zhang et al., 2011). d) AOM index (GDGT2/Crenarchaeol ratios) (Weijers et al., 2011). e) Water column overprint values (GDGT-2/GDGT-3 ratios) (Taylor et al., 2013). f) Methanogenesis values (GDGT-0/Crenarchaeol ratios) (Blaga et al., 2009).**

**Supplementary Figure S3: Cross plot between the ring index and $TEX_{86}$ values of samples from DSDP Site 274. The lines mark the outer ranges of the ring index (Zhang et al., 2016), outside of which samples have outlying values (marked as crosses). The shade of blue indicates the sample depth (mbsf).**



**Table 1**

| *In situ* protoperidioid taxa | Code | *In situ* gonyaulacoid taxa | Code |
|---|---|---|---|
| *Brigantedinium pynei* | *Br pyn* | *Achomosphaera alcicornu* | *Ac alc* |
| *Brigantedinium simplex* | *Br sim* | *Batiacasphaera* spp. pars | *Ba* spp |
| *Brigantedinium* spp. pars. | *Br spp* | *Batiacasphaera cooperi* | *Ba coo* |
| *Lejeunecysta* spp.pars | *L* spp | *Batiacasphaera compta* | *Ba com* |
| *Lejeunecysta acuminata* | *L acu* | *Batiacasphaera* sp. B sensu Bijl et al., 2018 | *Ba* spB |
| *Lejeunecysta adeliensis* | *L ade* | *Cerebrocysta* spp. | *Cer* spp |
| *Lejeunecysta attenuata* | *L att* | *Cleistosphaeridium* sp A. sensu Bijl et al., 2018 | *Cl* spA |
| *Lejeunecysta fallax* | *L fal* | *Corrudinium* spp. pars | *Co* spp |
| *Lejeunecysta katatonos* | *L kat* | *Corrudinium labradori* | *Co lab* |
| *Lejeunecysta rotunda* | *L rot* | *Gelatia inflata* | *G inf* |
| *Lejeunecysta* sp. A | *L* spA | *Hystrichokolpoma bullatum* | *Hy bul* |
| *Malvinia escutiana* | *M esc* | *impagidinium* cf *aculeatum* | *I acu* |
| *Protoperidinium* indet. | *Prot* | *Impagidinium cantabrigiense* | *I can* |
| *Selenopemphix antarctica* | *Se ant* | *Impagidinium velorum* | *I vel* |
| *Selenopemphix brinkhusii* | *Se bri* | *Impagidinium victorium* | *I vic* |
| *Selenopemphix nephroides* | *Se nep* | *Impagidinium paradoxum* | *I par* |
| *Selenopemphix* spp. pars | *Se spp* | *Impagidinium pallidum* | *I pal* |
| *Dinocyst* sp. 1 | Dino sp1 | *Impagidinium* sp. A sensu Bijl et al., 2018 | *I* spA |
| **Reworked peridinioid cysts** | | *Nematosphaeropsis labyrinthus* | *N lab* |
| *Alterbidinium distinctum* | *Al dis* | *Operculodinium* sp. A sensu Bijl et al., 2018 | *O* spA |
| *Deflandrea* spp. pars | *Df* spp | *Operculodinium centrocarpum* | *O cen* |
| *Moria zachosii* | *M zac* | *Operculodinium eirikianum* | *O eir* |
| *Phthanoperidinium* spp. pars | *Ph* spp | *Operculodinium janduchenei* | *O jan* |
| *Senegalinium* spp. | *Sen* spp | *Operculodinium piasekii* | *O pia* |
| *Spinidinium* spp. pars | *Spd* spp | *Operculodinium* spp. pars | *O* spp |
| *Vozzhennikovia* spp. pars | *Voz* spp | *Pyxidinopsis* spp. | *Pyx* spp |
| Other P-cyst reworked | otr-P | *Spiniferites ramous* | *Sf ram* |
| | | *Spiniferites bulloideus* | *Sf bul* |
| | | *Spiniferites* spp. pars | *Sf* spp |
| | | *Stoveracysta kakanuiensis* | *St kak* |
| | | *Stoveracysta ornata* | *St orn* |
| | | **Reworked gonyaulacoid cysts** | |
| | | *Arachnodinium antarcticum* | *A ant* |
| | | *Cerebrocysta* spp. pars RW | *Cer* RW |
| | | *Corrudinium regulare* | *Co reg* |
| | | *Corrudinium incompositum* | *Co inc* |
| **Other palynomorphs** | | *Enneadocysta* spp. pars | *Enn* spp |
| Unidentified Dinocyst 1 | Indet 1 | *Hystrichokolpoma rigaudiae* | *H rig* |
| Unidentified Dinocyst 2 | indet 2 | *Hystrichosphaeridium truswelliae* | *Hy tru* |
| Unidentified Dinocyst 3 | indet 3 | *Impagidinium* spp. pars RW | *I RW* |
| Terrestrial | Terr | *Operculodinium* spp. RW | *Ope RW* |
| Pterospermella/green algae | Ptero | *Pentadinium laticinctum* | *P lat* |
| Acritarch spp. | Acrit spp | *Thalassiphora pelagica* | *Th pel* |
| Acritarch chorate/spiney spp. | Acri spiney | *Tuberculodinium vancampoae* | *T van* |
| Leiosphaeridia | Leios | *Turbiosphaera* spp. pars RW | *Tur* spp |
| *Cymatosphaera* spp. pars | Cym Spp | Other G-cyst reworked | otr-G |





**Table 2**

| FO/LO/HO | Genus, Chron | Species | Age (Ma) | Depth | Depth error | Event source |
|---|---|---|---|---|---|---|
| | Base of C7n | | 24.47 | 181.23 | | This study |
| HO | *Chiasmolithus* | *altus* | 25.44 | 190.8 | | Burns, 1975; Kulhanek et al., 2019 |
| | Base of C8n.2n | | 25.99 | 198.74 | | This study |
| | Base of C9n | | 27.44 | 305.96 | | This study |
| LO | *Stoveracysta* | *ornata* | 30.8 | 323.655 | 2.015 | This study |
| FO | *Corrudinium* | *labradori* | 30.92 | 362.42 | 1.24 | This study |
| | Base of C12n | | 31.03 | 363.44 | | This study |
| FO | *Stoveracysta* | *ornata* | 32.5 | 396.62 | 5.25 | This study |
| | Base of C13n | | 33.7 | 400.17 | | This study |
| FO | *Malvinia* | *escutiana* | 33.7 | 404.66 | n/a | This study |





**Figure 1**

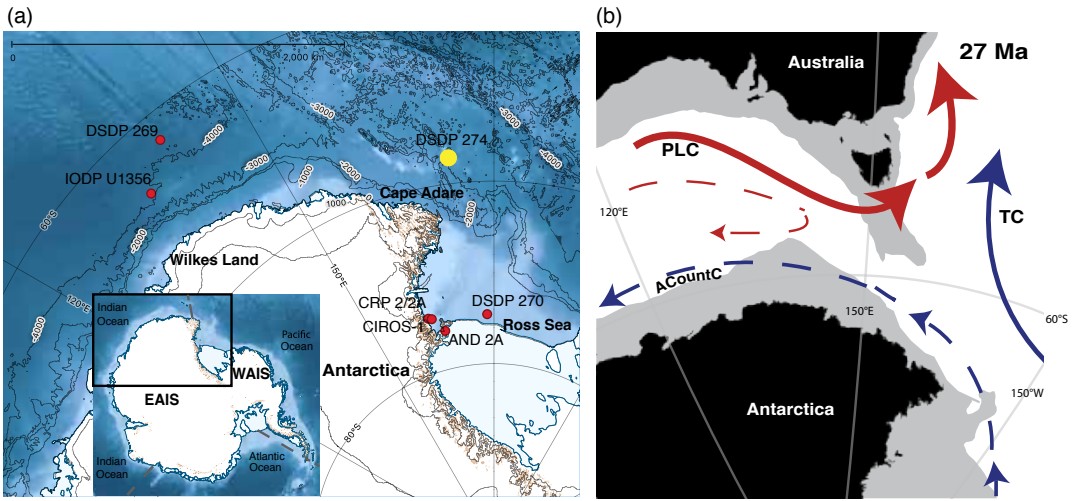



**Figure 2**

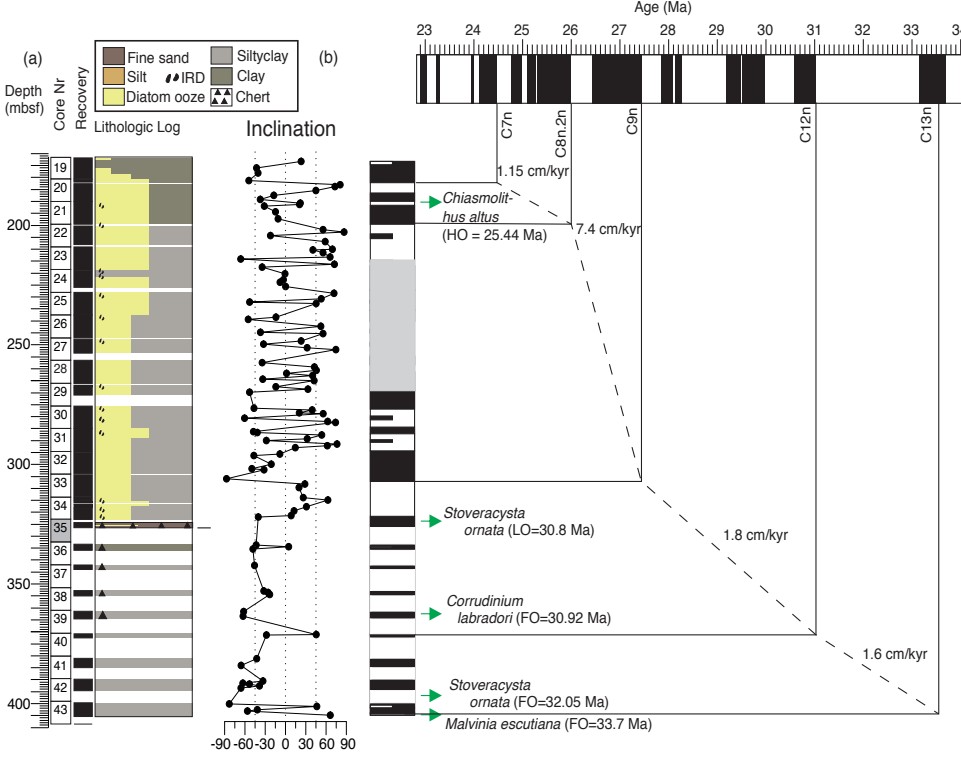



**Figure 3**

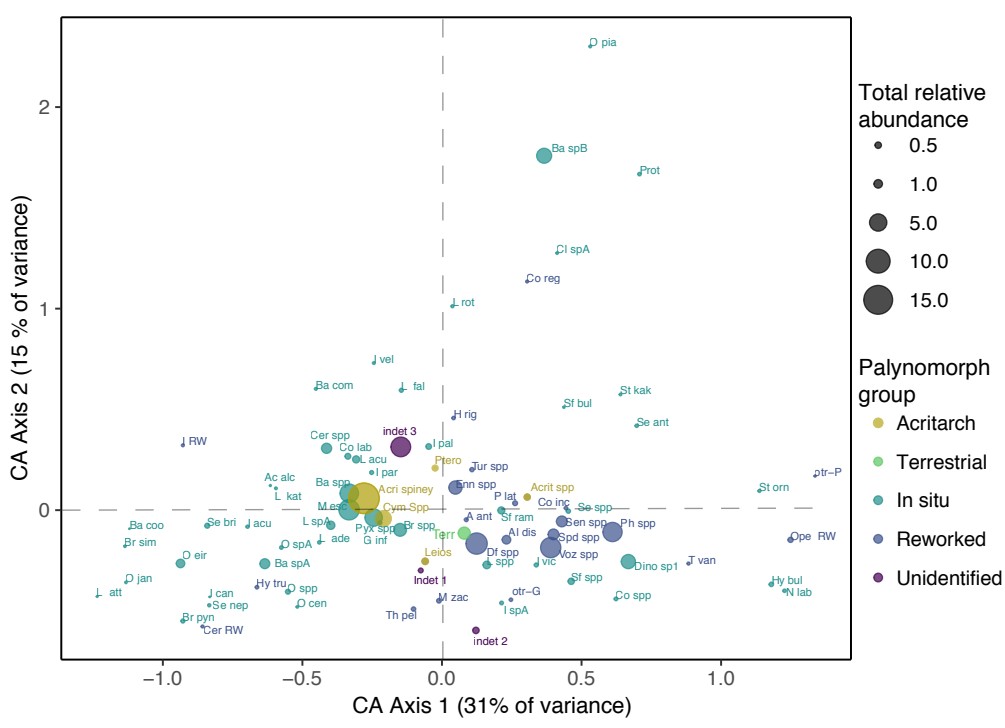

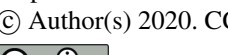



**Figure 4**

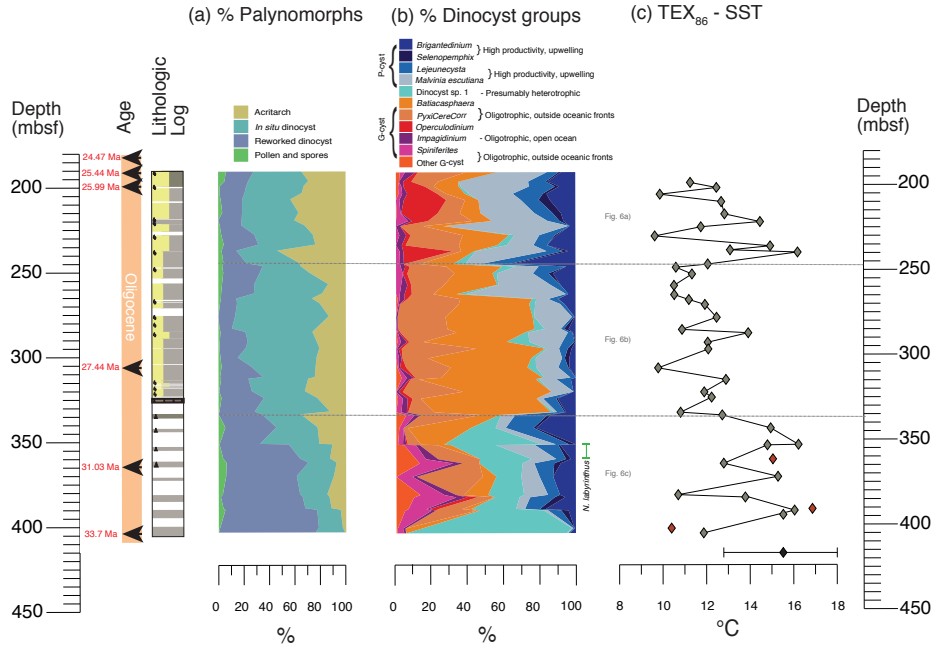





**Figure 5**

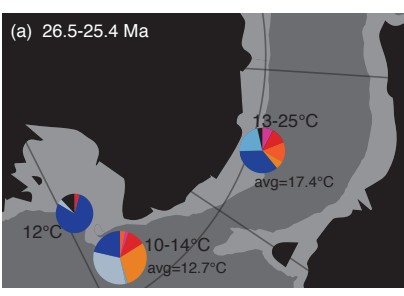

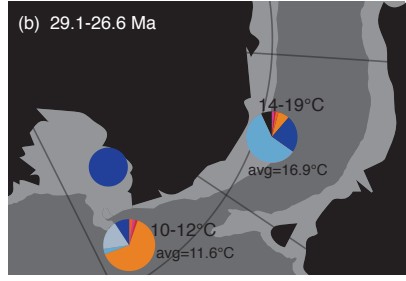

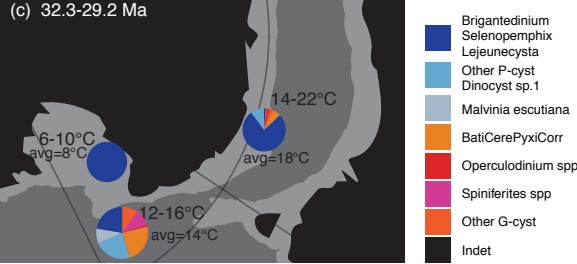