# Peer review of "Temperate Oligocene surface ocean conditions offshore Cape Adare, Ross Sea, Antarctica"

_Climate of the Past, 2020_

## Referee Comment (RC1) · Anonymous Referee #1 · 7 Jan 2021

Dear editor, authors,

Hoem et al., submitted a manuscript to Climate of the Past Discussions titled Temperate Oligocene surface ocean conditions offshore Cape Adare, Ross Sea, Antarctica. In this manuscript, the authors present newly generated gdgt, biostratigraphic, magnetostratigraphic and palynology data from DSDP Site 274, located offshore the Ross Sea continental margin, near the Antarctic continent. The records span the early Oligocene, are characterized by a relatively coarsely resolved bio-magneto age model, poorly preserved PMAG signals, relatively high reconstructed SSTs/temperate (sub)surface water conditions, and dinoflagellate cyst assemblies that vary in their composition throughout the three selected Oligocene intervals. These results are used for paleoclimatic, paleoceanographic and Antarctic cryosphere reconstructions. The text

is well written and the figures look good.

Given what is possible with old DSDP material and the patchy nature of records proximal to the Antarctic ice sheet, the authors have done a thorough job. The strongest point of this study are the reconstructed SSTs. I must admit that I am not very impressed by the bio-magneto stratigraphic age control. Also, the presented component analysis leaves more than half the variance unexplained. Yet, despite these limitations the authors provide a very extensive interpretation. I feel that in some instances the authors stack speculation on top of speculation, which may lead to tunnel-vision to make their results fit those from other sites and/or the literature.

Despite these issues, I recommend publication, mainly because as a paleoclimate community we need to work with whatever natural archives are available. However, I would like to ask the authors to address/rebut the issues I raise here. I leave it to the editor to decide how far the authors should go in adapting the manuscript (I do not need to see the manuscript again).

Major comments:

1: One of my major concerns is that the authors make a point about improving the age model, but show their results in the depth domain. With such an elaborate paleoclimatic interpretation in the discussion, I believe that the authors should show their results in the age domain. This also includes providing some age uncertainties.

2: I strongly recommend to delete or very strongly tone down section 5.4. To me this reads as a step too far in interpreting SST and dino results on a sketchy age model. Speculation stacked on speculation. What do the authors think dinos are/are not a proxy for? Alternatively, if the authors insist on keeping this section, then formulate some testable hypotheses based on these rather speculative interpretations.

3: I do not think the orbital part of the story can be backed up by the data.

Minor comments (per line number/section):

29, 30: "orbital states". This is rather vague. The records do not resolve any orbits, not even close to any orbital frequency. These interpretations thus rely on the analysis of Levy et al., based on the record of Pälike et al., and the authors attempt at making a coherent story out of both their own data and these previous results. In principle, this is an admirable effort, but I wonder if their data the author's present could also be interpreted to disagree with Levy et al., if an opposite phase relation to the orbits is found (if/when better age constraints become available)?

41, 42: "Oi-1". I suggest the author's follow Hutchinson et al. 2018, nomenclature for EO events. Oi-1 becomes EOIS.

47: "Oligocene". give approximate ages. The Oligocene is not one uniform time slice.

53: "still constricted". What is meant here. Surface connection only, no connection?

62: "colder temperatures". Choose: either colder waters, or lower temperatures, but not colder temperatures. Just like, you are heavy and weigh much, but cannot weigh heavy. Fix throughout.

64: perhaps not call it a gradient with the Tasman straight in between. Call it a temperature difference between the two sites.

64: add "(sub)surface" before "ocean temperature"

71: "before"? Do you mean "in front of"?

74: replace "classic" by "more standard inorganic geochemical"?

75: "strong links". This is the weak part of dino-based interpretations. Can you quantify these strong links? Those I find in this paper are rather weak! PCA-1 only explains 31%. A lot of interpretations and speculation are based on these strong links. I would tone this down.

79: any indication of the average age uncertainty/diachrony of bio events would be helpful here. How good are these constraints?

80: "improve the age model". Ok, but by how many depth-to-age tie-points was the age model improved. I just saw a paper come online by Jovane et al., 2020, who also improved the ages for Site 274. How do your new ages compare? Can you include their results and improve the age model further?

99: "diatom-rich". Could these not be used for dating?

117: "early" instead of "lower", because ages are given in brackets

121: Could rad or diatom stratigraphy help improve the age model further?

164: add….fucus when interpreting the results. Section 3.3.2.: I would move this to/integrate in the discussion.

219: delete "in", "general" should be "generally"

220-228: I would delete the PMAG results. They don't add much to the story or age model. Alternatively, integrate the new results by Jovane et al., 2020, to make more of these age constraints. Somewhere between 229 and 248: Explain how many depth to gts2012 age tie-points are added compared to the previous shipboard age model.

289: Sentence starting with "Given that…" needs a citation.

291: Dinocyst sp. 1 could not have been reported previously, as it is defined in this study

293: "argues for in situ production" Why? Does ocean transport break down cysts?

299: absence of evidence is not evidence of absence. Does the high g-cyst abundance not argue for the opposite?

307: replace "temperature" with "gdgts". This is still the results section. Leave the interpretations for the discussion.

Section 4.3.4. Potential delete. These results don't add much, and are not really discussed.

318: 46% is rather low in my opinion. Together with the rather poor age control, this is the main issue with this data set. No very clear signals.

324: given the low variance explained, I think "gives confidence" is a bit of an over statement. I understand that the authors attempt to sell their data, but I would stick with the facts. ... The data isn't very clear. And much more data is needed to test many of the hypotheses and interpretations of the authors.

331: by how many tie-points? With respect to what?

339: "..., a period...600 ppm". This reads like a tag on to the rest of the sentence. Make the link to CO2 relevant/explicit.

342: Can the authors briefly explain/recap to non-dino folk what P-cysts and G-cysts are 'proxies' for in general/this setting/according to the literature?

350/351: I understand that G-cysts are proxies for upwelling and oligotrophic conditions. Ok. How well-established is this (perhaps give some modern-day r2 values, PCA percentages between these cysts and processes)? Or refer to the literature that makes this interpretation obvious/standard/accepted.

359: to my reading the interpretation that the Eocene dinocysts are reworked is largely/only based on the interpreted, rather poorly resolved age model. Are there no alternative explanations possible? Could the age model be wrong and the Eocene in situ? If not, why not? Make explicit.

Section 5.2.2; What is the (average) sampling resolution in the age domain? Do the authors still feel confident to go orbital with their interpretations? To what level? 405 kyr eccentricity? 100 kyr? Or even 40 kyr obliquity and beyond? My feeling is that the data is not suitable for such astronomical interpretations later on in the manuscript, and would advice the authors to stick with comparing/contrasting Oligocene climate "states". "Orbital states" is not a thing, because the system never equilibrates to the relatively short lasting/high frequency orbital configurations.

369: start of new section. Remove "also".

371/385: replace "proto...noid" with P-cyst. This has already been abbreviated.

386-389: surely gdgts are more easily reworked that dinocysts. Yet the authors argue the opposite.

390: "high temperatures", not "warm temperatures". Alternatively, "warm sub surface waters".

392: can you provide an R2 for this covariance?

399-400: remains rather arm-wavy. Can some of these relationships be quantified?

405-406. G-cysts are relatively more abundant because of a decrease in P-cysts. Make the effects of such a closed sum effect on interpreting your data explicit.

Section: 5.3: This reads like a review paper. Quite a lot of speculation based on limited data. I guess this is the nature of the game, but you might lose the attention of some readers when several levels of speculation are stacked. Perhaps be more cautious?

437: "ocean crustal connection". Do the authors mean a deep-water connection?

462: "we can now exclude. ....". Remind me again, why that is?

465: "related to orbital cyclicity". This is a very vague statement. What orbit? The data presented cannot support this.

471: Perhaps mention winnowing as a reason too?

480: Regarding the point about heightened obliquity sensitivity. This is solely based on the Levy interpretation of the partially obliquity tuned Pälike data. I understand that this fits your interpretation and may give context to understanding the results from Site 274, but the newly presented data cannot confirm or refute or support the Levy hypothesis. I would make this point.

487: Could winnowing have removed diatoms and dinos?

529: "precession driven top down melting". Pls remove. There is no data presented to support this statement.

533: When? During the entire Oligocene?

538: "gradient". The authors argue that a gradient does not make much sense with a Tasmanian gateway in between these sites.

541: Sentence starting with "During cold phases, . . ." I can see how this argument works for heat, but for moist you'd expect the opposite. Again, I would refrain from using dinos to interpret cryospheric conditions.

I wish to congratulate the authors with a well written and nicely illustrated paper.

References:

Hutchinson et al., The Eocene-Oligocene transition: a review of marine andterrestrial proxy data, models and model-data comparisons, Climate of the Past, 2020, https://cp.copernicus.org/preprints/cp-2020-68/

Jovane et al., Magnetostratigraphic Chronology of a Cenozoic Sequence From DSDP Site 274, Ross Sea, Antarctica, Frontiers in Earth Science, 2020, https://www.researchgate.net/profile/Luigi_Jovane/publication /345947592_ Magnetostratigraphic_Chronology_of_a_Cenozoic_ Sequence_From_DSDP_Site_274_Ross_Sea_Antarctica /links/5fd787d7a6fdccdcb8c58a88 /Magnetostratigraphic- Chronology-of-a-Cenozoic- Sequence-From-DSDP-Site-274- Ross-Sea-Antarctica.pdf

---

## Referee Comment (RC2) · Anonymous Referee #2 · 8 Mar 2021

The manuscript of Hoem et al, fills in an important gap in the understanding of the Oligocene Antarctic ice sheet. The results, interpreted as evidence for relatively warm SST's offshore Antarctica during the Oligocene, are consistent with the lack of ice-rafted debris from the Wilkes Land core. Overall, I find the manuscript compelling, and my comments are limited to minor revisions, with the exception of a comment about the age model. The manuscript is well structured and organized. My one criticism are some awkward turns of phrase, that could be remedied easily during the pre-publication phase.

Line 33: awkward wording, "from warm influence from. . .

Line 114-115: Sentence is a fragment. Also do not begin a sentence with a numerical symbol (e.g. 200. . .)

Line 225: "central part of the site", change to "upper Oligocene section of the core"

Line 324 and elsewhere: latin phrases like a priori should be italicized

Line 358-359: change to "the region could have been under the influence of. . ."

Line 428-429: awkward sentence structure, suggest rewording

Line 451: mid-Oligocene is not a recognized stratigraphic interval. Maybe say "latest early Oligocene to earliest late Oligocene".

Age model

There is considerable uncertainty in the age determinations for the early Oligocene. For example the age model datums indicate ∼400 m/Myr between the ornata and labradori datums.

Would the authors also please comment and justify the assignment of the normal magnetozone in core 40 to C12n? I also note the tie line between the normal magnetozone in core 40 and C12n is incorrectly placed (discussed below in the figures comments).

Overall, the discussion of the uncertainty in the age model is honest and realistic.

Figures

Fig 2

I suggest plotting the age of the biostrat datums as well as indicating the depth.

The tie line between the reversal boundary at approx. 373 m and the base of C12n has been incorrectly placed. The line tied to the base of C12n cannot be tied to the top of a normal chron in the magstrat record.

―――――――――――――――――――――

---

## Author Comment (AC1) · 12 Apr 2021

Dear referee and editorial team, Here below, you find our response to the comments the reviewer rose on our paper Hoem et al., CP- 2020-139. We thank the reviewer for their thorough, constructive and positive feedback on our manuscript. In general, we fully understand some of the concerns and mostly agree with the referees' comments. We propose the changes indicated in the text below, which will have no impact on the results and conclusions we drew in the submitted version.

Best regards, Frida S. Hoem on behalf of all co-authors

Response to Reviewer #1 Dear editor, authors, Hoem et al., submitted a manuscript to

Climate of the Past Discussions titled Temperate Oligocene surface ocean conditions offshore Cape Adare, Ross Sea, Antarctica. In this manuscript, the authors present newly generated gdgt, biostratigraphic, magnetostratigraphic and palynology data from DSDP Site 274, located offshore the Ross Sea continental margin, near the Antarctic continent. The records span the early Oligocene, are characterized by a relatively coarsely resolved bio-magneto age model, poorly preserved PMAG signals, relatively high reconstructed SSTs/temperate (sub)surface water conditions, and dinoflagellate cyst assemblies that vary in their com- position throughout the three selected Oligocene intervals. These results are used for paleoclimatic, paleoceanographic and Antarctic cryosphere reconstructions. The text is well written and the figures look good. Given what is possible with old DSDP material and the patchy nature of records proximal to the Antarctic ice sheet, the authors have done a thorough job. The strongest point of this study are the reconstructed SSTs. I must admit that I am not very impressed by the bio-magneto stratigraphic age control. Also, the presented component analysis leaves more than half the variance unexplained. Yet, despite these limitations the authors provide a very extensive interpretation. I feel that in some instances the authors stack speculation on top of speculation, which may lead to tunnel-vision to make their results fit those from other sites and/or the literature. Despite these issues, I recommend pub- lication, mainly because as a paleoclimate community we need to work with whatever natural archives are available. However, I would like to ask the authors to address/rebut the issues I raise here. I leave it to the editor to decide how far the authors should go in adapting the manuscript (I do not need to see the manuscript again).

Authors response: We thank the referee for these constructive comments and will respond in detail below.

Major comments: Comment #1: One of my major concerns is that the authors make a point about improving the age model, but show their results in the depth domain. With such an elaborate paleoclimatic interpretation in the discussion, I believe that the authors should show their results in the age domain. This also includes providing some

age uncertainties.

RESPONSE: We indeed revisit and improve the age model, as we used revised ages of the ocean crust underneath the site, and added new paleomagnetic and dinocyst biostratigraphic constraints. However, because the age model is still poor (DSDP holes have poor recovery), we preferred to present the dinocyst assemblage data into depth domain, while keeping the tentative age indicated with red text and arrows (Fig. 4). The full reasoning of our decision is explained between lines 333 and 337. Presenting the data in the age domain would unjustifiably give the impression of continuity of the record and robust age control. That said, we do understand the need for the community to potentially use our temperature (SST) and other data for comparison to other records. For this reason, the data will be made available in both depth and age domain (Table S2 and S3).

PROPOSED CHANGES: We propose to leave the data in depth domain in the Figure, while the text (mostly) presents data both in depth and in age and we present the data both in age and in depth in the supplementary dataset, using our age model. This allows users to compare results among different sites and allows recalculation of ages (based on depth) should the age model further improve.

Comment #2: I strongly recommend to delete or very strongly tone down section 5.4. To me this reads as a step too far in interpreting SST and dino results on a sketchy age model. Speculation stacked on speculation. What do the authors think dinos are/are not a proxy for? Alternatively, if the authors insist on keeping this section, then formulate some testable hypotheses based on these rather speculative interpretations.

RESPONSE: Section 5.4 tries to reconcile the warmth seen at Site 274 with the overwhelming evidence of a glaciated Oligocene Antarctic continent, while embedding our results within the existing literature. Notably, more proximal sites are inferring warm ocean conditions during the Oligocene. Independently from the robustness of the age model, and even if we had analyzed the warmest (interglacial) intervals only, the reconstructed warm conditions at DSDP 274 and other proximal sites is a fact. Although we agree that the implications of the warmth are not yet fully understood, we disagree with the reviewer that this lack of understanding is due to the poor age constraints. Our low-resolution age model exclusively prevents us from making detailed one-to-one time comparisons with other records, which we did not do. Our age model only allows comparisons with other records with same ages as those for which we have control. In section 5.4, we merely place the warmth throughout our Oligocene record into perspective of other records spanning the Oligocene. This comparison forces us to rethink how a warm Oligocene Southern Ocean can coexist with a sizeable Antarctic ice sheet that has marine terminations. Dinocysts are an often-used proxy for temperature, nutrients and productivity, sea-ice and salinity reconstructions. Dinocysts (as other biological-based proxies) are extensively studied in the modern ocean so that we know which of the environmental variable drives the spatial distribution of each species. Moving back in time, certainties in the relation between one species and the major driver of its distribution obviously decrease, either because some species are extinct or because their ecological niche may have changed partly. Our dinocyst-based oceanographic implications are based on a model we have clearly outlined in the methods (section 3.3.2 and references therein), objectively presented in the results (section 4.3) and interpreted in the discussion (section 5.2). All of this is common practice. We followed a testable hypothesis to derive our interpretations. Next to the extant species/groups for which the ecological interpretation may be easier, the statistical analyses, which includes also the extinct species is a way to make the interpretation more solid. We do appreciate the suggestion to add testable hypotheses to section 5.4, which could direct future research efforts.

PROPOSED CHANGES: We will carefully reconsider the speculative nature of section 5.4 particularly with the age constraints in mind. We will add testable hypotheses to section 5.4.

Comment #3: I do not think the orbital part of the story can be backed up by the data.

RESPONSE: We never intended to suggest that our record has a high enough resolution to resolve orbital frequencies; in fact we carefully avoided this. As the reviewer suggests, it is indeed the other way around: we try to reconcile the high variability in our reconstruction by ascribing it to the strong variability in polar climate on orbital time scales, as was demonstrated in other studies (e.g., Levy et al., 2019), irrespective of whether this is obliquity, precession or eccentricity. This follows the same basic approach as in the trilogy on the Oligocene record at IODP Site U1356 (Salabarnada et al., 2018; Bijl et al., 2018; Hartman et al., 2018).

PROPOSED CHANGES: see below at the specific comments

Minor comments (per line number/section):

29, 30: "orbital states". This is rather vague. The records do not resolve any orbits, not even close to any orbital frequency. These interpretations thus rely on the analysis of Levy et al., based on the record of PaÌĹlike et al., and the authors attempt at making a coherent story out of both their own data and these previous results. In principle, this is an admirable effort, but I wonder if their data the author's present could also be interpreted to disagree with Levy et al., if an opposite phase relation to the orbits is found (if/when better age constraints become available)?

RESPONSE: Here, we correctly state that we reconcile the strong variability in our record by the observations from other studies that show strong orbital variability in specific time intervals in the Oligocene.

41, 42: "Oi-1". I suggest the author's follow Hutchinson et al. 2018, nomenclature for EO events. Oi-1 becomes EOIS.

RESPONSE: Correct, we will change the nomenclature accordingly.

47: "Oligocene". give approximate ages. The Oligocene is not one uniform time slice.

RESPONSE: We will describe the time span of the Oligocene once, in the beginning of the introduction, and add which geologic time scale we apply (Gradstein et al., 2012).

53: "still constricted". What is meant here. Surface connection only, no connection?

RESPONSE: We will add "narrow" to the sentence to clarify.

62: "colder temperatures". Choose: either colder waters, or lower temperatures, but not colder temperatures. Just like, you are heavy and weigh much, but cannot weigh heavy. Fix throughout.

RESPONSE: We will change the text accordingly, and check throughout the manuscript for consistency.

64: perhaps not call it a gradient with the Tasman straight in between. Call it a temperature difference between the two sites.

RESPONSE: We will change the text accordingly

64: add "(sub)surface" before "ocean temperature"

RESPONSE: We will change the text accordingly, and check throughout

71: "before"? Do you mean "in front of"?

RESPONSE: the right word is "between"

74: replace "classic" by "more standard inorganic geochemical"?

RESPONSE: We will change the text accordingly

75: "strong links". This is the weak part of dino-based interpretations. Can you quantify these strong links? Those I find in this paper are rather weak! PCA-1 only explains 31%. A lot of interpretations and speculation are based on these strong links. I would tone this down.

RESPONSE: it is clear that our text is creating confusion. We are here referring to the research done on surface samples at how dinocyst link to modern conditions very well (Zonneveld et al., 2013; Prebble et al., 2013). We will rephrase and refer to these citations.

79: any indication of the average age uncertainty/diachrony of bio events would be helpful here. How good are these constraints?

RESPONSE: yes, this is an appropriate remark. We will further elaborate on the uncertainties of the biostratigraphy in subchapter 4.1

80: "improve the age model". Ok, but by how many depth-to-age tie-points was the age model improved. I just saw a paper come online by Jovane et al., 2020, who also improved the ages for Site 274. How do your new ages compare? Can you include their results and improve the age model further?

RESPONSE: A dinocyst-based biostratigraphy and new magnetostratigraphic constraints integrated with available biostratigraphic data are used to improve the model. We provide the age model with nine new tie-points. These may be few extra data points, nonetheless they represent an improvement.

PROPOSED CHANGES: We will include the number of new age constraints. We will also add the magnetostratigraphic constraints from Jovane et al., 2020 to our age model figure for comparison between the two paleomagnetic data (See updated version of Figure 2 at the bottom of the rebuttal letter).

99: "diatom-rich". Could these not be used for dating?

RESPONSE: Diatom (and rads) biostratigraphy at Site 274 is unfortunately not available; it could indeed improve the age model in the future.

117: "early" instead of "lower", because ages are given in brackets

RESPONSE: We will change the text accordingly, and check throughout

121: Could rad or diatom stratigraphy help improve the age model further? RESPONSE: See above comments (99) on diatoms

164: add....fucus when interpreting the results.

RESPONSE: We will change the text accordingly.

Section 3.3.2.: I would move this to/integrate in the discussion.

RESPONSE: We think this is really a subjective decision, not really impacting the quality of the work. Our choice of explaining the relationship between paleoceanography and dinocysts in the methods, rather than in the discussion, is meant to make the discussion more focused on the implications and general environmental and paleocenographic interpretations rather than interrupting the reasoning flow with details on the proxy itself.

PROPOSED CHANGES: We will clarify in section 3.3.2 the method of using existing, and established models which utilizes the modern relationship between dinocysts with properties of the overlying water (e.g Prebble et al., 2013), and the relationship between dinocysts and other paleoceanographic proxies for temperature, runoff/fresh water input, and nutrient conditions (Frieling and Sluijs, 2018) to infer paleoceanographic conditions from the palynological assemblages. A clearer explanation of how we use existing "models" for paleoceanographic relationship of dinocysts, can also help alleviate some of the concerns the reviewer is raising below in line 342 and 350/351.

219: delete "in", "general" should be "generally"

RESPONSE: We will change the text accordingly.

220-228: I would delete the PMAG results. They don't add much to the story or age model. Alternatively, integrate the new results by Jovane et al., 2020, to make more of these age constraints.

PROPOSED CHANGES: We will add a comparison with the Jovane et al., 2020 in Figure 2 (see bottom of letter) comparing the respective paleomagnetic results and by adding this text to section 4.1: "Magnetostratigraphic results for the upper Oligocene are similar than those produced by Jovane et al., (2020), who carried out a thorough study of the magnetic properties. For the lower Oligocene, our biostratigraphic results provide new tie-points, which indicate lower Oligocene ages rather than an upper Eocene age as in Jovane et al. (2020) and in the initial reports."

Somewhere between 229 and 248: Explain how many depth to gts2012 age tie-points are added compared to the previous shipboard age model.

RESPONSE We will add this to the text accordingly.

289: Sentence starting with "Given that. . ." needs a citation.

RESPONSE We will add the citation of Sluijs et al., (2005) accordingly.

291: Dinocyst sp. 1 could not have been reported previously, as it is defined in this Study

RESPONSE We will add "Informally named" Dinocyst sp. 1, as it is the first time this dinocyst is reported. It is here not defined (i.e., formally described) and as far as we know never reported in previous studies even under another name.

293: "argues for in situ production" Why? Does ocean transport break down cysts?

RESPONSE: Not necessarily. Reworking and down slope slumping exposes the dinocysts to degradation, which could be visible in the preservation state of the cysts. Generally, well preserved cysts suggest that dinocysts are produced/deposited in situ.

299: absence of evidence is not evidence of absence. Does the high g-cyst abundance not argue for the opposite?

RESPONSE: In this case it does. Despite high g-cyst abundance, the presence of p-cysts argues against selective preservation (P cysts are more sensitive than G cysts to degradation due to oxic condition). Furthermore, the lithology, abundance of amorphous organic matter and preservation of biomarkers argues against strongly oxygenated bottom conditions.

PROPOSED CHANGES: We will rephrase this part to make the point of no changes in

lithology are observed that would explain the decrease in relative P-cyst appearance (Zonneveld et al., 2010)

307: replace "temperature" with "gdgts". This is still the results section. Leave the interpretations for the discussion.

RESPONSE: We will change "temperature" to "TEX86-SST "

Section 4.3.4. Potential delete. These results don't add much, and are not really discussed.

RESPONSE: We argue that this is still a valuable observation.

318: 46% is rather low in my opinion. Together with the rather poor age control, this is the main issue with this data set. No very clear signals.

RESPONSE: Given the diversity of the palynomorph assemblages, with 80 taxa, having 2 axes explain almost half the variance is in our view an indication that meaningful relationships exist between axis scores and environmental parameters.

PROPOSED CHANGES: We will add to the paper that this 46% is good, given the high dinocyst diversity.

324: given the low variance explained, I think "gives confidence" is a bit of an over statement. I understand that the authors attempt to sell their data, but I would stick with the facts. . .. The data isn't very clear. And much more data is needed to test many of the hypotheses and interpretations of the authors.

RESPONSE: We argue, given the multidimensionality of the dinocyst data, with many taxa, the result of the CA is actually quite good. Moreover, the positions of the taxa in the 2-dimensional space of the CA is consistent with inference based on our knowledge of modern cyst affinities and empirical data.

331: by how many tie-points? With respect to what?

RESPONSE: We will add the number of extra age tie points added by our data.

339: ". . ., a period. . .600 ppm". This reads like a tag on to the rest of the sentence. Make the link to CO2 relevant/explicit.

RESPONSE: We will remove the link to CO2 in this paragraph

342: Can the authors briefly explain/recap to non-dino folk what P-cysts and G-cysts are 'proxies' for in general/this setting/according to the literature?

RESPONSE: This was explained in detail in "3.3.2 Dinocyst paleoecological affinity".

PROPOSED CHANGES: To make it clearer we will add "nutrient rich conditions" after P-cyst and "oligotrophic conditions" after G-cyst were already mentioned.

350/351: I understand that G-cysts are proxies for upwelling and oligotrophic conditions. Ok. How well-established is this (perhaps give some modern-day r2 values, PCA percentages between these cysts and processes)? Or refer to the literature that makes this interpretation obvious/standard/accepted.

RESPONSE: We need to correct the reviewer here. In reality, in the modern ocean, G-cysts are usually indicative of oligotrophic water and not of upwelling. P-cysts are mostly indicators of high nutrients conditions and they are very abundant in upwelling area. In chapter 3.3.2. Dinocyst paleoecological affinity, we lay out the method behind inferring paleoecological properties through studying dinocyst assemblages, by utilizing existing and established models. We there refer to literature such as Prebble et al. (2013) and Zonneveld et al., (2013) explaining the modern-day affinities of dinocysts, and Esper and Zonneveld (2007), Frieling and Sluijs (2018) explain the paleoecological affinities of dinocysts.

359: to my reading the interpretation that the Eocene dinocysts are reworked is largely/only based on the interpreted, rather poorly resolved age model. Are there no alternative explanations possible? Could the age model be wrong and the Eocene in situ? If not, why not? Make explicit.

RESPONSE: We interpret the lowermost sediment package at Site 274 to be no older than early Oligocene both because of the basalt ocean crust which was dated to chron 13 by Cande et al., 2000, and because of the presence of Malvinia escutiana at the base of the section. The first occurrence of Malvinia escutiana is at 33.7 Ma (Bijl et al., 2018a; Houben et al., 2011). Hence, the Early Oligocene age of the bottom sediment is robust despite abundance of typical Eocene dinocysts. However, the last occurrence of these typical Eocene taxa is subject to debate and might vary between areas. Species that have a last occurrence at the EOT continue into the early Oligocene at some sedimentary sections. Although in these sections it is argued that these are reworked due to the 50 m lower sea level at EOT (ÅŽliwińska et al., 2019), proving this is complicated. Bijl et al. (2018b) attempted this by looking at covariance of assumed reworked species abundance, to distinguish reworked from in situ taxa. We argue that the regional geography at Site 274 is likely susceptible for reworking from the continental shelf, in line with the processes at Site U1356 by Bijl et al. (2018b).

PROPOSED CHANGES: We will explain this better in this section.

Section 5.2.2; What is the (average) sampling resolution in the age domain? Do the authors still feel confident to go orbital with their interpretations? To what level? 405 kyr eccentricity? 100 kyr? Or even 40 kyr obliquity and beyond? My feeling is that the data is not suitable for such astronomical interpretations later on in the manuscript, and would advice the authors to stick with comparing/contrasting Oligocene climate "states". "Orbital states" is not a thing, because the system never equilibrates to the relatively short lasting/high frequency orbital configurations.

RESPONSE: The resolution of the data is indeed of too low resolution to fully capture orbital cyclicity. Rather we argue the other way around: we explain the high variability in our record as forced by the strong orbital-induced climate/oceanographic variability that is expected in this polar setting. Indeed, our resolution cannot exactly pinpoint which orbital parameter causes the variability, and we therefore avoided to do this. We merely point to a study (Levy et al., 2019, drawing mostly from the Ross Sea records) from

nearby locations where similar high orbital response of the environmental conditions is reported. Moreover, the high variability is in line with interpretations from the Wilkes Land Margin, where orbital cyclicity in the lithology was demonstrated.

369: start of new section. Remove "also".

RESPONSE: We will change the text accordingly

371/385: replace "proto. . .noid" with P-cyst. This has already been abbreviated.

RESPONSE: We will change the text accordingly

386-389: surely gdgts are more easily reworked that dinocysts. Yet the authors argue the opposite.

RESPONSE: We do not know whether GDGTs are more easily reworked than dinocysts, we are unaware of any studies which report this. We do assume a weaker preservation potential for GDGTs than for dinocysts, particularly when transported, as lipid biomarkers are more susceptible to oxic degradation and maturation than sporopollenin/dinosporin. Moreover, if a part of the GDGTs were reworked (from more inland) then this would be reflected by high BIT index values, which is very low throughout the record.

390: "high temperatures", not "warm temperatures". Alternatively, "warm sub surface waters".

RESPONSE: We will change the text accordingly

392: can you provide an R2 for this covariance? 399-400: remains rather arm-wavy. Can some of these relationships be quantified? RESPONSE: We surmise that the comments of the reviewer on the speculative nature of our interpretations from the data stem from the inadequate description of our model to reconstruct paleoceanography from dinocysts.

PROPOSED CHANGES: We will explain this model more explicitly in the methods

section.

405-406. G-cysts are relatively more abundant because of a decrease in P-cysts. Make the effects of such a closed sum effect on interpreting your data explicit.

RESPONSE: Yes, we argue that the relative amount of G-cyst increase, as a consequence of a lack of nutrients causing the relative %p-cyst to go down (Esper and Zonneveld, 2007).

PROPOSED CHANGES: We will change the last sentence to: "..., which leaves the remaining G-cysts higher in the relative abundance of total dinocyst assemblages" to clarify. Additionally, we will site Esper and Zonneveld (2007), who demonstrate that the %p-cyst is a proxy for nutrient conditions in the Southern Ocean.

Section: 5.3: This reads like a review paper. Quite a lot of speculation based on limited data. I guess this is the nature of the game, but you might lose the attention of some readers when several levels of speculation are stacked. Perhaps be more cautious?

RESPONSE: In section 5.3, we place the paleoceanographic reconstruction for Site 274, into context of other paleoceanographic and paleocryospheric reconstructions in the region. Specifically, we try to reconcile marine-terminating ice sheets in the Ross Sea area (Levy et al., 2019) with warm ocean conditions and high climate variability on orbital time scales (our study, Hartman et al., 2018 and Levy et al., 2019). We do not see how this can be too speculative, and without further specification of which aspect the reviewer finds too speculative, we cannot propose any adjustments.

437: "ocean crustal connection". Do the authors mean a deep-water connection?

RESPONSE: We agree and will change the sentence accordingly to "deep-water connection"

462: "we can now exclude. . ...". Remind me again, why that is?

RESPONSE: The indices for overprint in TEX86 values are laid out in chapter 3.2.2,

the results of the potential non-thermal values are shown in Fig. S2 and presented in chapter 4.2.

PROPOSED CHANGES: We will add a reference to Fig S2, where the potential biases are presented.

465: "related to orbital cyclicity". This is a very vague statement. What orbit? The data presented cannot support this.

RESPONSE: We will remove "related to orbital cyclicity" from the sentence.

471: Perhaps mention winnowing as a reason too?

RESPONSE: Stronger ocean bottom currents could cause the oxic conditions we proposed was the reason behind the disappearance of dinocysts. Winnowing would not selectively erode palynomorphs away and would result in coarsening of sediments, which we do not see. The lithological examination of the 192.7 - 181 mbsf interval where dinocyst are barren, show diatom rich silty-clay. We therefore don't believe that winnowing can explain the disappearance dinocysts. PROPOSED CHANGES: We will add this possibility to the manuscript 480: Regarding the point about heightened obliquity sensitivity. This is solely based on the Levy interpretation of the partially obliquity tuned PaÌĹlike data. I understand that this fits your interpretation and may give context to understanding the results from Site 274, but the newly presented data cannot confirm or refute or support the Levy hypothesis. I would make this point.

RESPONSE: There is no sentence in section 5.3 or 5.4 in which we claim to prove or disprove the concept of Obliquity sensitivity of Levy et al., 2019. We merely stratigraphically correlate a period with strong orbital amplitude in the ANDRILL record (and Wilkes Land) to the same strong variability in our record, demonstrating high-amplitude environmental variability in the system. This coincides with a time interval of strong variability in the benthic foraminiferal $\delta$18O, which is indicative of either large ice volume changes (i.e., an ice sheet sensitive to climate forcing) or high variability in deep-sea

temperature (which in essence reflects polar SST changes). Indeed, this does not say anything about the exact orbital parameter that is at play here, but that is not what we argue.

487: Could winnowing have removed diatoms and dinos? RESPONSE: See comment above to line 471.

529: "precession driven top down melting". Pls remove. There is no data presented to support this statement.

RESPONSE: We will remove the extra information from the text

533: When? During the entire Oligocene?

RESPONSE: We will add the age of the study interval (33.7 – 24.5 Ma) to clarify.

538: "gradient". The authors argue that a gradient does not make much sense with a Tasmanian gateway in between these sites.

RESPONSE: We here refer to the temperature difference between warmer sea surface conditions at Site 274 offshore the Ross Sea Margin and the colder sea surface conditions inshore the ross sea. Thus, in this case the Tasmanian Gateway is not between the inner and outer Ross Sea.

PROPOSED CHANGES: We will change the word "gradient" with "difference".

541: Sentence starting with "During cold phases, ..." I can see how this argument works for heat, but for moist you'd expect the opposite. Again, I would refrain from using dinos to interpret cryospheric conditions.

RESPONSE: The reviewer is right that this is confusing. We meant to say that the overall warmer climates and warmer Southern Ocean (in glacial or interglacial phases) in general increases precipitation and thus ice accumulation, relative to the dryness of present-day. The glacial-interglacial cyclicity creates variability to that precipitation and temperature, but the large overall precipitation flux ensures mass balance of the ice

sheet during warmer-than-present-day climates.

PROPOSED CHANGES: We will cite Speelman et al. (2010), who showed evidence for stronger poleward moisture transport that is less depleted (i.e., from a more local source) during warmer climates.

I wish to congratulate the authors with a well written and nicely illustrated paper. We thank the reviewer again for this useful discussion

References: Hutchinson et al., The Eocene-Oligocene transition: a review of marine andterres- trial proxy data, models and model-data comparisons, Climate of the Past, 2020, https://cp.copernicus.org/preprints/cp-2020-68/

Jovane et al., Magnetostratigraphic Chronology of a Cenozoic Sequence From DSDP Site 274, Ross Sea, Antarctica, Frontiers in Earth Science, 2020, https://www.researchgate.net/profile/Luigi_Jovane/publication /345947592_ Magnetostratigraphic_Chronology_of_a_Cenozoic_ Sequence_From_DSDP_Site_274_Ross_Sea_Antarctica /links/5fd787d7a6fdccdcb8c58a88 /Magnetostratigraphic- Chronology-of-a-Cenozoic- Sequence-From-DSDP-Site-274- Ross-Sea-Antarctica.pdf

References

Bijl, P. K., Houben, A. J., Bruls, A., Pross, J., and Sangiorgi, F.: Stratigraphic calibration of Oligocene-Miocene organic-walled dinoflagellate cysts from offshore Wilkes Land, East Antarctica, and a zonation proposal, Journal of Micropalaeontology, 37, 105-138, 2018a. Bijl, P. K., Houben, A. J., Hartman, J. D., Pross, J., Salabarnada, A., Escutia, C., and Sangiorgi, F.: Paleoceanography and ice sheet variability offshore Wilkes Land, Antarctica-Part 2: Insights from Oligocene-Miocene dinoflagellate cyst assemblages, Climate of the Past, 14, 1015-1033, 2018b. Esper, O. and Zonneveld, K. A.: The potential of organic-walled dinoflagellate cysts for the reconstruction of past sea-surface conditions in the Southern Ocean, Marine Micropaleontology, 65, 185-212, 2007. Frieling, J. and Sluijs, A.: Towards quantitative environmental reconstructions from ancient non-analogue microfossil assemblages: Ecological preferences of Paleocene–Eocene dinoflagellates, Earth-Science Reviews, 185, 956-973, 2018. Houben, A. J., Bijl, P. K., Guerstein, G. R., Sluijs, A., and Brinkhuis, H.: Malvinia escutiana, a new biostratigraphically important Oligocene dinoflagellate cyst from the Southern Ocean, Review of Palaeobotany and Palynology, 165, 175-182, 2011. Jovane, L., Florindo, F., Wilson, G., Leone, S. d. A. P. S., Hassan, M. B., Rodelli, D., and Cortese, G.: Magnetostratigraphic Chronology of a Cenozoic Sequence From DSDP Site 274, Ross Sea, Antarctica, Multi-Disciplinary Applications in Magnetic Chronostratigraphy, 2020. 2020. ÅŽliwińska, K. K., Thomsen, E., Schouten, S., Schoon, P. L., and Heilmann-Clausen, C.: Climate-and gateway-driven cooling of Late Eocene to earliest Oligocene sea surface temperatures in the North Sea Basin, Scientific reports, 9, 1-11, 2019. Speelman, E. N., Sewall, J. O., Noone, D., Huber, M., von der Heydt, A., Damsté, J. S., and Reichart, G.-J.: Modeling the influence of a reduced equator-to-pole sea surface temperature gradient on the distribution of water isotopes in the Early/Middle Eocene, Earth and Planetary Science Letters, 298, 57-65, 2010. Zonneveld, K. A. F., Versteegh, G. J. M., Kasten, S., Eglinton, T. I., Emeis, K. C., Huguet, C., Koch, B. P., de Lange, G. J., de Leeuw, J. W., Middelburg, J. J., Mollenhauer, G., Prahl, F. G., Rethemeyer, J., and Wakeham, S. G.: Selective preservation of organic matter in marine environments; processes and impact on the sedimentary record, Biogeosciences, 7, 483-511, 2010.

———————————————

[Figure]

**Fig. 1.**

---

## Author Comment (AC2) · 12 Apr 2021

Dear referee and editorial team,

Here below, you find our response to the comments the reviewer rose on our paper Hoem et al., CP- 2020-139. We thank the reviewer for constructive and positive feedback on our manuscript. We propose the changes indicated in the text below.

Best regards, Frida S. Hoem on behalf of all co-authors

Response to anonymous Referee #2

The manuscript of Hoem et al, fills in an important gap in the understanding of the Oligocene Antarctic ice sheet. The results, interpreted as evidence for relatively warm

SST's offshore Antarctica during the Oligocene, are consistent with the lack of ice-rafted debris from the Wilkes Land core. Overall, I find the manuscript compelling, and my comments are limited to minor revisions, with the exception of a comment about the age model. The manuscript is well structured and organized. My one criticism are some awkward turns of phrase, that could be remedied easily during the pre-publication phase.

Authors response: We thank the reviewer for the positive opinion of the paper, and will respond to the minor revisions accordingly below.

Line 33: awkward wording, "from warm influence from. . .

PROPOSED CHANGES: We will change this to: "the influence of warmer water"

Line 114-115: Sentence is a fragment. Also do not begin a sentence with a numerical symbol (e.g. 200. . .)

RESPONSE: We will change the text accordingly

Line 225: "central part of the site", change to "upper Oligocene section of the core"

RESPONSE: We will change the text accordingly

Line 324 and elsewhere: latin phrases like a priori should be italicized

RESPONSE: We follow the CP submission guidelines, which states: "Common Latin phrases are not italicized"

Line 358-359: change to "the region could have been under the influence of. . ." RESPONSE: We will change the text accordingly

Line 428-429: awkward sentence structure, suggest rewording

RESPONSE: We agree and will change the wording

Line 451: mid-Oligocene is not a recognized stratigraphic interval. Maybe say "latest early Oligocene to earliest late Oligocene".

RESPONSE: We will change the text accordingly

Age model There is considerable uncertainty in the age determinations for the early Oligocene. For example the age model datums indicate âĹij400 m/Myr between the ornata and labradori datums.

RESPONSE: Yes, there is uncertainty in the age model, last occurrence datums (in the case of S. ornata) are less robust than first occurrence datums due to the fact that species can get reworked and deposited in sediments younger than when they lived.

Would the authors also please comment and justify the assignment of the normal magnetozone in core 40 to C12n? I also note the tie line between the normal magnetozone in core 40 and C12n is incorrectly placed (discussed below in the figures comments).

RESPONSE: We agree on this. The change of polarity, which we suggest it may correspond to the change between C12r and C12n, needs to be placed above, in the core 39.

PROPOSED CHANGES: Accordingly, we will move this line to core 39 as indicated above. Overall, the discussion of the uncertainty in the age model is honest and realistic.

Figures Fig 2 I suggest plotting the age of the biostrat datums as well as indicating the depth.

RESPONSE: The top panel of the figure indicates the age. We argue that having the age constraints written behind the biostratigraphic markers together with the lines drawn between the paleomagnetic chrons and their respective GTS2012 age is sufficient.

PROPOSED CHANGES: In the age model Figure 2 (see attached figure below) we will incorporate the paleomagnetic data of (Jovane et al., 2020) to come to a state-of-the-art age model reconstruction. The tie line between the reversal boundary at approx. 373 m and the base of C12n has been incorrectly placed. The line tied to the base of

C12n cannot be tied to the top of a normal chron in the magstrat record.

RESPONSE: We thank the reviewer's observation, and we will move the line in Figure 2

References: Jovane, L., Florindo, F., Wilson, G., Leone, S. d. A. P. S., Hassan, M. B., Rodelli, D., and Cortese, G.: Magnetostratigraphic Chronology of a Cenozoic Sequence From DSDP Site 274, Ross Sea, Antarctica, Multi-Disciplinary Applications in Magnetic Chronostratigraphy, 2020. 2020.

[Figure]

**Fig. 1.**